# CODiff: One-Step Diffusion Model for Camouflaged Object Detection

Xiaotong Fu [1]   Qian Liu [2]   Qihang Zhou [1]   Wenchao Meng [1]   Qinmin Yang [1]   Shibo He [1]

## Abstract

Diffusion-based camouflaged object detection (COD) has recently shown great potential. In contrast to existing approaches that rely on multiple sample steps to refine the predicted masks, we propose CODiff, which reformulates the diffusion process to enable one-step mask prediction while maintaining competitive accuracy. Specifically, we first establish the theoretical feasibility of one-step sampling for COD. Based on this, we design a dedicated network for one-step inference with a global semantic guidance mechanism to guide the denoising process globally and hierarchical condition integration blocks to provide fine-grained structural semantics. In addition, we design a straight-forward regularization to learn better intermediate features by bridging the representation gap between the condition backbone and the diffusion model. Extensive experiments demonstrate that CODiff achieves state-of-the-art performance across multiple benchmarks, improving MAE by over 22% on the challenging COD10K dataset. Code is available at https://github.com/KiiSooo/CODiff.

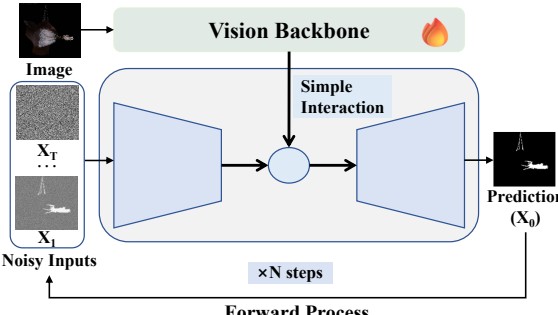

*(a)* Previous methods require multi-steps to refine the prediction mask.

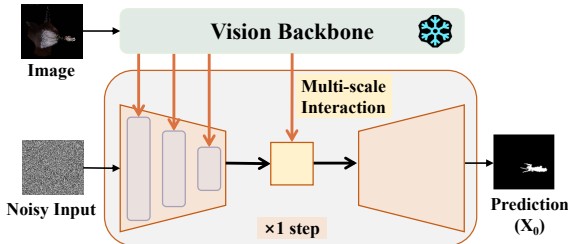

*(b)* Our method achieves outperforming performance with one-step inference.

*Figure 1.* Comparison between our model and previous diffusion-based COD methods.

## 1. Introduction

Camouflaged Object Detection (COD) aims to identify objects that blend seamlessly into the background, which has significant applications in various practical fields, such as industrial defect detection (Kumar, 2008) and medical image segmentation (Li et al., 2022; Rahman et al., 2023). However, camouflaged objects share highly similar patterns with their surrounding environment, such as edges, textures, or colors, which makes COD a valuable yet challenging visual task.

[1]College of Control Science and Engineering, Zhejiang University, Hangzhou, China [2]School of Mathematics and Statistics, Xi'an Jiaotong University, Xi'an, China. Correspondence to: Wenchao Meng <wmengzju@zju.edu.cn>.

*Proceedings of the 43rd International Conference on Machine Learning*, Seoul, South Korea. PMLR 306, 2026. Copyright 2026 by the author(s).

Previous COD models are mainly built upon CNNs (Fan et al., 2022; He et al., 2023; Ji et al., 2023) or Transformers (Hu et al., 2023; Huang et al., 2023; Xing et al., 2023; Yin et al., 2024; Zhang et al., 2025). Although these architectures have achieved encouraging results, they adopt a discriminative learning paradigm, directly regressing a single deterministic prediction for the target mask, which limits their ability to model the underlying distribution of camouflaged objects. In contrast, diffusion models formulate the learning process from a generative perspective and are capable of approximating data distributions through iterative denoising. Recently, diffusion models have achieved remarkable success in image generation (Rombach et al., 2022; Peebles & Xie, 2023; Ma et al., 2024; Wang et al., 2025). Beyond generation, the denoising process enables diffusion models to capture meaningful semantic representations (Kwon et al., 2023; Xiang et al., 2023; Yu et al., 2025), which has inspired their extension to discriminative vision tasks such as object detection (Chen et al., 2023a)

and semantic segmentation (Baranchuk et al., 2022; Qu et al., 2025). For the COD task, it can be reformulated as a diffusion-based mask generation problem and demonstrates promising performance (Chen et al., 2024), which suggests that modeling mask distribution via diffusion models is a promising direction.

However, existing diffusion-based COD models (Chen et al., 2023b; 2024; Yang et al., 2025) inherit the conventional diffusion paradigm, treating COD as a complex stochastic generation process. This perspective overlooks a fundamental characteristic of the task: the conditional distribution of camouflaged object masks is inherently low-complexity and highly constrained. Consequently, these methods resort to multi-step sampling to progressively refine predictions, introducing unnecessary sampling steps.

Specifically, by formulating COD as a temporally iterative process, prior works rely on uncertainty modeling or multi-step ensemble strategies to compensate for insufficient modeling capacity, which necessitates multiple inference steps. Moreover, the capacity of such models is often restricted by simplistic conditioning mechanisms such as concatenation (Fig. 1a), which fail to fully exploit the fine-grained structural features essential for locating complex camouflaged boundaries. Finally, existing approaches ignore the representation gap between the frozen conditional backbone and the diffusion model, resulting in extra training for the vision backbone.

To overcome these limitations, we propose CODiff, a novel one-step diffusion-based framework for COD. We first analyze the feasibility of one-step generation in COD, demonstrating that the inherent low-complexity of the COD task allows for accurate approximation within a single step, provided the model architecture possesses sufficient representation capacity. To realize this potential, we introduce a Global Semantic Guidance (GSG) mechanism that globally guides the denoising process with both semantic and temporal information, facilitating global object-aware guidance for COD. Furthermore, Hierarchical Condition Integration (HCI) modules are designed to enable deep interaction between the diffusion model's intermediate features and multi-scale condition features, ensuring precise localization of complex boundaries and small objects. Finally, recognizing that the representation gap between the condition encoder and the diffusion model will limit convergence and performance, we propose a Representation Alignment Loss ($\mathcal{L}_{RA}$) as a straightforward regularization term.

In summary, our model benefits from a novel guidance mechanism, multi-scale condition interaction, and aligned semantic intermediate representations to achieve superior COD performance in a single-step reverse process. Extensive experiments on multiple benchmark datasets demonstrate that our CODiff significantly outperforms the SOTA approaches.

To the best of our knowledge, this is the first work to address the COD task via a one-step diffusion model. The main contributions of this paper are summarized as follows:

- **Rethinking COD as One-step Generation**. Different from previous multi-step methods, we exploit the intrinsic characteristics of the COD task and formulate it as a single-step diffusion problem.

- **Architecture Design**. We propose a novel diffusion-based architecture tailored for one-step COD, which effectively integrates multi-level image features and diffusion timestep information to provide hierarchical semantic and structural guidance.

- **Empirical Validation**. Empirical evaluation on multiple COD benchmarks demonstrates that CODiff achieves SOTA performance, highlighting the feasibility and potential of one-step diffusion for structured prediction tasks.

## 2. Related Work

### 2.1. Classical Models for COD

Traditional approaches, including CNN-based models (Li et al., 2021; Mei et al., 2021; Sun et al., 2021; Yang et al., 2021; Zhai et al., 2021) and Transformer-based models (Hu et al., 2023; Lyu et al., 2023; Xing et al., 2023; Yin et al., 2024), have proposed a variety of solutions for the COD task from different perspectives. Several methods attempt to capture subtle discriminative features through frequency information (Zhong et al., 2022), texture modeling (Zhai et al., 2022), or contextual features (Chen et al., 2022; Zhou et al., 2024). Other approaches are inspired by the visual perception mechanism of camouflaged objects, introducing various strategies such as focusing, zooming (Pang et al., 2022; 2024), and shrinking (Jia et al., 2022) to effectively locate camouflaged objects. Compared with CNN-based models, vision Transformers can model long-range dependencies more effectively through attention mechanisms. Notable Transformer-based methods include FSPNet (Huang et al., 2023), which introduces a non-local mechanism, HitNet (Hu et al., 2023), which employs feedback refinement on low-resolution representations, and CamoFormer (Yin et al., 2024), which progressively refines decoding results. In addition, several semi-supervised (Yan et al., 2025b) and unsupervised (Du et al., 2025b; Yan et al., 2025a) methods have been explored for the COD task, showing promising results.

However, most traditional methods formulate COD as a regression problem, producing a single deterministic prediction by optimizing pixel-wise probabilities, which is insufficient to capture the inherent ambiguity of camouflaged scenes. Therefore, we reformulate COD as a distribution

learning problem and design a diffusion model to learn the joint distribution of segmentation masks.

## 2.2. Diffusion Models for COD

In recent years, diffusion models have achieved remarkable success in various generative tasks, including image and video synthesis (Saharia et al., 2022; Rombach et al., 2022; Peebles & Xie, 2023; Ma et al., 2024; Labs et al., 2025). Since diffusion models learn data distributions and generate data that follow the same distribution, a growing number of studies have tried to extend the scope of diffusion models beyond synthesis, exploring their potential in discriminative tasks. Several works (Brempong et al., 2022; Baranchuk et al., 2022) have demonstrated that diffusion models are architecturally similar to dense prediction models, making their representations a strong candidate for discriminative tasks such as instance segmentation (Gu et al., 2024), semantic segmentation (Brempong et al., 2022; Baranchuk et al., 2022), medical image segmentation (Wu et al., 2024; Rahman et al., 2023; Lin et al., 2024), and both supervised (Chen et al., 2023b; 2024; Yang et al., 2025) and unsupervised (Du et al., 2025a) camouflaged object detection. Diffusion-based COD methods corrupt the object masks with noise and generate predictions by conditioning denoising steps with prior features extracted by vision backbones, enabling the model to learn the distribution of camouflaged objects. These methods typically integrate the conditional feature into the UNet bottleneck of the diffusion model by concatenation and convolution.

However, existing diffusion-based COD approaches primarily adopt traditional diffusion paradigms. Such designs do not account for the inherent low-complexity of the target mask distribution in COD, resulting in requiring multi-step sampling or specific ensemble strategies to refine the prediction due to insufficient conditioning and denoising guidance. In contrast, our proposed method, CODiff, revisits the diffusion model for COD from a task-oriented perspective, focusing on the theoretical formulation and practical realization of a one-step diffusion-based model with outperforming performance.

## 3. Preliminaries

We present a brief overview of diffusion models (Ho et al., 2020; Song et al., 2021), which model the target distribution $p(x)$ via learning a gradual denoising process from a Gaussian distribution $\mathcal{N}(\mathbf{0}, \mathbf{I})$ to $p(x)$. Formally, diffusion models learn a reverse process $p(x_{t-1} \mid x_t)$ of the pre-defined forward process $q(x_t|x_0)$ that gradually adds the Gaussian noise starting from $p(x)$ for $1 \leq t \leq T$ with a fixed $T > 0$. For a given $x_0 \sim p(x)$, $q(x_0|x_{t-1})$ can be formalized as $q(x_t \mid x_0) = \mathcal{N}(x_t; \sqrt{\bar{\alpha}_t}x_0, (1 - \bar{\alpha}_t)\mathbf{I})$, where $\beta_t \in (0, 1)$ are pre-defined small hyperparameters.

In particular, DDPM (Ho et al., 2020) shows if one formalizes the reverse process $p(x_{t-1}|x_t)$ (with $\alpha_t = 1 - \beta_t$, $\bar{\alpha}_t = \prod_{i=1}^{t} \alpha_i$ for $1 \leq t \leq T$) as :

$$\mu_\theta = \frac{1}{\sqrt{\alpha_t}}\Big(x_t - \frac{\beta_t}{\sqrt{1 - \bar{\alpha}_t}}\boldsymbol{\epsilon_\theta}(x_t, t)\Big),$$
$$p_\theta(x_{t-1}|x_t) = \mathcal{N}\Big(x_{t-1}; \ \mu_\theta, \boldsymbol{\Sigma_\theta}(x_t, t)\Big), \tag{1}$$

where $\boldsymbol{\Sigma_\theta}(x_t, t)$ is simply defined as $\sigma_t^2 \mathbf{I}$ with $\beta_t = \sigma_t^2$. $\boldsymbol{\epsilon_\theta}(x_t, t)$ is trained as follows :

$$\mathcal{L}_{\text{simple}} = \mathbb{E}_{x_0, \boldsymbol{\epsilon}, t}\Big[||\boldsymbol{\epsilon} - \boldsymbol{\epsilon_\theta}(x_t, t)||_2^2\Big]. \tag{2}$$

Despite DDPM, previous work (Song et al., 2021) has formulated this problem via a Stochastic Differential Equation (SDE) and derives a continuous-time reverse process as follows :

$$dx_t = \Big(-\frac{1}{2}\beta(t)x_t - \beta(t)\nabla_x \log q(x_t)\Big)dt + \sqrt{\beta(t)}dw, \tag{3}$$

where w is a Wiener process.

The deterministic generative Probability Flow (PF) Ordinary Differential Equation (ODE) can be formulated as :

$$dx_t = -\frac{1}{2}\beta(t)\big(x_t - \nabla_x \log q(x_t)\big)dt. \tag{4}$$

## 4. Methodology

### 4.1. Theoretical Formulation for One-Step COD

We propose CODiff to generate a probabilistic mask $x_0$ from given RGB images $\mathbf{I}$ that contains camouflaged objects through a one-step reverse process. Firstly, by integrating the PF ODE from time $t$ to 0 starting from $x_t$ (see Appendix A for details), we have:

$$\mathbf{x}_0 - \mathbf{x}_t = -\int_t^0 \frac{1}{2}\beta(s)\left[\mathbf{x}_s - \nabla_{x_s} \log q_s(\mathbf{x}_s)\right]ds. \tag{5}$$

Then we introduce $\mathbf{F}(\mathbf{x}_t, t, \mathbf{I})$ to represent the right-hand side of this equation and we can reformulate the above equation as :

$$\mathbf{x}_0 - \mathbf{x}_t = -\mathbf{F}(\mathbf{x}_t, t, \mathbf{I}) \implies \mathbf{x}_0 = \mathbf{x}_t - \mathbf{F}(\mathbf{x}_t, t, \mathbf{I}) \tag{6}$$

According to Eq. (15) in Appendix A, we can generate an accurate $\mathbf{x}_0$ by calculating $\mathbf{F}(\mathbf{x}_t, t, \mathbf{I})$ once. In this work, we define a neural network-parameterized function $\mathbf{F}_{\boldsymbol{\theta}}(\mathbf{x}_t, t, \mathbf{I})$ and $\hat{\mathbf{x}}_0 = \mathbf{x}_t - \mathbf{F}_{\boldsymbol{\theta}}(\mathbf{x}_t, t, \mathbf{I})$. Thus, it is necessary to ensure $\mathbf{F}_{\boldsymbol{\theta}}(\mathbf{x}_t, t, \mathbf{I}) \xrightarrow{\mathcal{L}} \mathbf{F}(\mathbf{x}_t, t, \mathbf{I})$ to achieve good COD performance.

**Definition 4.1.**

$$\mathcal{L}_{\text{total}}(\boldsymbol{\theta}) := \mathbb{E}_{t \sim \mathcal{U}[0,T]} [\mathbb{E}_{\mathbf{x}_0 \sim p_{\text{data}}(\mathbf{x}_0)}$$
$$[\mathbb{E}_{\mathbf{x}_t \sim \mathcal{N}(\sqrt{\bar{\alpha}_t}\mathbf{x}_0,(1-\bar{\alpha}_t)\mathbf{I})}$$
$$[\mathbb{E}_{\mathbf{I} \sim p_{\text{img}}(\mathbf{I})}$$
$$[d(\hat{\mathbf{x}}_0, \mathbf{x}_0)]]]],$$

where $\mathcal{U}[0,T]$ denotes a uniform distribution from time 0 (clean) to T (noisy). $d(\cdot, \cdot)$ is a metric function that satisfies for all pairs $(x, y)$, $d(x, y) \geq 0$, and $d(x, y) = 0$ if and only if $x = y$. The specific $\mathcal{L}_{\text{total}}$ we design will be discussed in Section 4.3.

This definition characterizes the expected discrepancy between the estimated $\hat{x}_0$ and the ground-truth (gt) $x_0$, integrated over the underlying data distribution and the time domain.

## 4.2. Convergence Analysis

We provide theoretical justifications for the convergence of out one-step CODiff framework as follows.

**Assumption 4.2.** The network-parameterized mapping $\mathbf{F}_{\boldsymbol{\theta}}(\mathbf{x}_t, t, \mathbf{I})$ is uniformly bounded and Lipschitz continuous with respect to $\mathbf{x}_t$, i.e. ,

$$\|\mathbf{F}_{\boldsymbol{\theta}}(\mathbf{x}_t, t, \mathbf{I})\| \leq B,$$
$$\|\mathbf{F}_{\boldsymbol{\theta}}(\mathbf{x}_t, t, \mathbf{I}) - \mathbf{F}_{\boldsymbol{\theta}}(\mathbf{x}'_t, t, \mathbf{I})\| \leq L\|\mathbf{x}_t - \mathbf{x}'_t\|.$$

for all $\mathbf{x}_t, \mathbf{x}'_t, t \in [0, T]$, and $\mathbf{I} \sim p_{\text{img}}(\mathbf{I})$.

**Theorem 4.3.** *If the training objective* $\mathcal{L}_{\text{total}}(\boldsymbol{\theta}) \to 0$, *then for any* $\epsilon > 0$,

$$\mathbb{P}\Big(d(\hat{\mathbf{x}}_0, \mathbf{x}_0) > \epsilon\Big) \to 0 \quad as \quad \mathcal{L}_{\text{total}}(\boldsymbol{\theta}) \to 0,$$

*where* $d(\cdot, \cdot)$ *is the metric satisfying Definition 4.1. Consequently,*

$$\mathbf{F}_{\boldsymbol{\theta}}(\mathbf{x}_t, t, \mathbf{I}) \xrightarrow{\mathbb{P}} \mathbf{F}(\mathbf{x}_t, t, \mathbf{I}),$$

*i.e., the learned mapping converges in probability to the true reverse mapping induced by the diffusion process.*

*Proof.* As $\mathcal{L}_{\text{total}}(\boldsymbol{\theta}) \to 0$, we have :

$$\mathbb{E}_{t,\mathbf{x}_0,\mathbf{x}_t,\mathbf{I}}\big[d(\hat{\mathbf{x}}_0, \mathbf{x}_0)\big] \to 0.$$

Using Markov's inequality, for any $\epsilon > 0$ ,

$$\mathbb{P}\big(d(\hat{\mathbf{x}}_0, \mathbf{x}_0) > \epsilon\big) \leq \frac{\mathbb{E}[d(\hat{\mathbf{x}}_0, \mathbf{x}_0)]}{\epsilon} \to 0.$$

This establishes convergence in probability :

$$\hat{\mathbf{x}}_0 \xrightarrow{\mathbb{P}} \mathbf{x}_0.$$

According to the definition in Section 4.1 and Assumption 4.2, we have:

$$\mathbf{F}_{\boldsymbol{\theta}}(\mathbf{x}_t, t, \mathbf{I}) \xrightarrow{\mathbb{P}} \mathbf{F}(\mathbf{x}_t, t, \mathbf{I}).$$

$\square$

**Assumption 4.4.** In the COD scenario, the gt mask $\mathbf{x}_0$ is nearly deterministic conditioned on the input image $\mathbf{I}$. Specifically, the conditional distribution $p_{\text{data}}(\mathbf{x}_0 \mid \mathbf{I})$ exhibits extremely low uncertainty, where the remaining variation mainly originates from annotation noise rather than semantic ambiguity. Formally, there exists a sufficiently small constant $\epsilon \approx 0$ such that:

$$\text{Var}[\mathbf{x}_0 \mid \mathbf{I}] \leq \epsilon.$$

**Corollary 4.5.** *Under Assumptions 4.2, 4.4 and Theorem 4.3, the following statement holds:*

$$\mathbb{E}\left[d\left(\hat{\mathbf{x}}_0, \mathbf{x}_0\right)\right] \leq \mathcal{L}_{\text{total}}(\theta) \to 0.$$

*Consequently,* $\hat{\mathbf{x}}_0$ *converges to* $\mathbf{x}_0$ *in expectation.*

Corollary 4.5 establishes that $\hat{\mathbf{x}}_0$ is statistically consistent with $\mathbf{x}_0$. From a distributional perspective, this implies that the push-forward distribution induced by the model, $p_\theta(\hat{\mathbf{x}}_0 \mid \mathbf{I})$, converges to the data distribution $p_{\text{data}}(\mathbf{x}_0 \mid \mathbf{I})$, which is sharply concentrated around the gt under Assumption 4.4.

Overall, the above arguments show that, under sufficient model capacity and the near-deterministic nature of COD, a one-step diffusion model can effectively learn the conditional distribution of camouflaged object masks.

### 4.3. Model Architecture

Guided by our theoretical formulation, we propose a specialized architecture that instantiates the neural network function $\mathbf{F}_{\boldsymbol{\theta}}(\mathbf{x}_t, t, \mathbf{I})$ as illustrated in Fig. 2. Since the network function $\mathbf{F}_{\boldsymbol{\theta}}(\mathbf{x}_t, t, \mathbf{I})$ takes three inputs $(\mathbf{x}_t, t, \mathbf{I})$, where $\mathbf{x}_t$ depends both on the target $\mathbf{x}_0$ and the timestep $t$, we design three components to fully exploit the information provided by the inputs.

**Global Semantic Guidance**. To introduce time awareness and global semantic information into the diffusion model, we design a GSG mechanism to globally guide the denoising process. We deploy it in all the attention blocks, including Self-Attention (SA) blocks and HCI blocks, as well as the final mask prediction head.

Specifically, we adopt the self-supervised vision model DINOv2 (Oquab et al., 2024) as the feature extractor, since it provides an explicit `<cls>` token that encodes global semantic information of the input image. GSG obtains several parameters including $(\alpha_1, \beta_1, \gamma_1)$ and $(\alpha_2, \beta_2, \gamma_2)$, through a simple network with MLP, SiLU and a linear layer, and

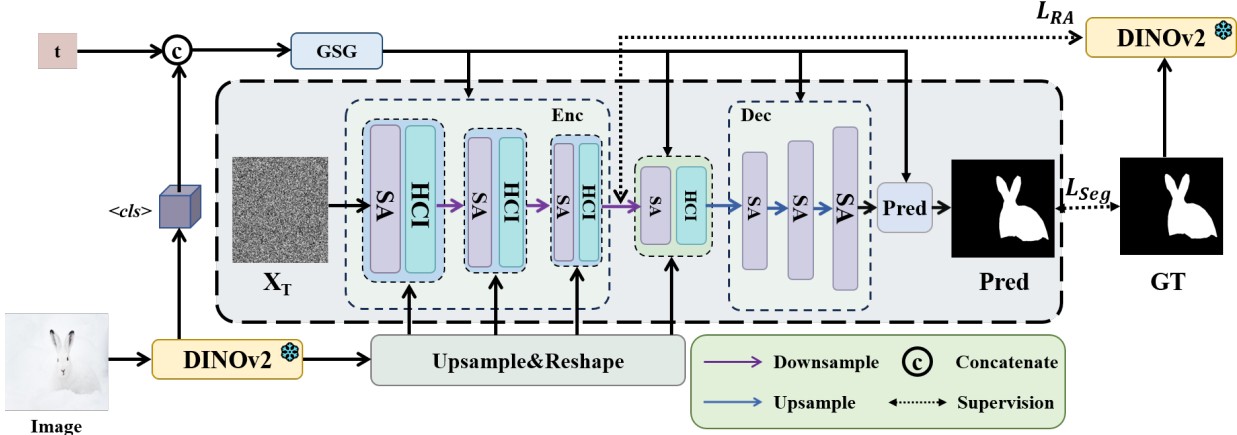

*Figure 2.* The main framework of our model. Our framework consists of a Global Semantic Guidance (GSG) mechanism, Hierarchical Condition Integration (HCI) blocks, and a representation alignment loss $L_{RA}$ (see Section 4.3 for details).

applies the parameters in the SA and HCI blocks. Given an input $x_{in}$ for an SA block, we have:

$$\begin{aligned} \text{mid} &= \gamma_1 \cdot \text{SA} \left( \alpha_1 \cdot x_{in} + \beta_1 \right) \\ x_{out} &= \gamma_2 \cdot \text{FF} \left( \alpha_2 \cdot (\text{mid} + x_{in}) + \beta_2 \right) \end{aligned} \quad (7)$$

where $\text{SA}(\cdot)$ and $\text{FF}(\cdot)$ denote the self-attention module and the feed-forward network, respectively.

As for the final prediction, we have:

$$\text{pred} = \text{Sigmoid} \left( \text{Conv} \left( (1 + \alpha_f) \cdot \text{LN}(x) + \beta_f \right) \right) \quad (8)$$

where $\text{Conv}(\cdot)$ and $\text{LN}(\cdot)$ denote a convolution layer and a layer normalization, respectively. The modulation parameters $\alpha_f$ and $\beta_f$ are produced by the GSG module and applied to the normalized feature for the final prediction.

The GSG mechanism can effectively integrate global semantic and temporal information in a computationally-efficient way, which ensures a semantic-aware and temporal-aware denoising process.

**Hierarchical Condition Integration**. To locate camouflaged objects with complex shapes or small sizes, we design HCI blocks to enable fine-grained conditioning by incorporating multi-level features.

We construct the query $Q_x$ from the input feature $x_{in}$ and the key-value pairs $KV_{cond}$ from the condition feature $\text{cond}$, and the parameters calculated by GSG are applied as follows:

$$\begin{aligned} \text{mid} &= \gamma_1 \cdot \text{CA} \left( KV_{cond}, \alpha_1 \cdot Q_x + \beta_1 \right) \\ x_{out} &= \gamma_2 \cdot \text{FF} \left( \alpha_2 \cdot (\text{mid} + x_{in}) + \beta_2 \right) \end{aligned} \quad (9)$$

where $\text{CA}(\cdot)$ and $\text{FF}(\cdot)$ denote the cross-attention module and the feed-forward network, respectively. We obtain multi-level condition features $\{F_i^I\}_{i=1}^4$ from the 4 selected layers

of DINOv2. These features are adjusted to the resolution of UNet features and employed as follows:

$$\begin{aligned} x_{in} &= \{\{E_i\}_{i=1}^3, h\} \\ \text{cond} &= \{F_i^I\}_{i=1}^4 \end{aligned} \quad (10)$$

where $\{E_i\}_{i=1}^3$ represent the feature maps from the three encoder stages of the UNet, and $h$ denotes the latent feature. The corresponding downsampling factors of the UNet encoder are 4, 2, and 2.

**Representation Alignment Loss**. To facilitate stable and effective intermediate representation learning, we propose a Representation Alignment Loss ($\mathcal{L}_{RA}$) as an explicit regularization term, which alleviates the semantic gap between the diffusion model and the vision backbone.

Additionally, researchers have validated that the encoder output feature in diffusion models contains high-level semantic information and has discriminative properties.(Kwon et al., 2023; Xiang et al., 2023). Hence, we employ $\mathcal{L}_{RA}$ to explicitly align the representations, which can be formulated as follows:

$$\mathcal{L}_{RA} = 1 - cossim(E_3, F_4^{gt}) \quad (11)$$

where $E_3, F_4^{gt}$ are from the UNet encoder and the vision backbone given clean masks, respectively, and $\text{cossim}(\cdot)$ denotes the cosine similarity function. $\mathcal{L}_{RA}$ mitigates the representation gap, which allows our method to utilize a frozen backbone and learn intermediate representations more efficiently.

Finally, the total loss $\mathcal{L}_{total}$ can be defined as follows:

$$\begin{aligned} \mathcal{L}_{total} &= \mathcal{L}_{seg} + \lambda \cdot \mathcal{L}_{RA} \\ \mathcal{L}_{seg} &= \mathcal{L}_{\omega BCE} + \mathcal{L}_{\omega IOU} + \mathcal{L}_{SSIM} + \mathcal{L}_{MAE}. \end{aligned} \quad (12)$$

*Table 1.* Quantitative comparison of three representative COD datasets. "↑" / "↓": higher/lower is better. The best, second-best, and third-best results are **bolded**, underlined, and *italicized*, respectively.

| Method | CAMO | | | | COD10K | | | | NC4K | | | | NFEs |
|---|---|---|---|---|---|---|---|---|---|---|---|---|---|
| | $S_\alpha\uparrow$ | $E_\phi\uparrow$ | $F_\beta^\omega\uparrow$ | M↓ | $S_\alpha\uparrow$ | $E_\phi\uparrow$ | $F_\beta^\omega\uparrow$ | M↓ | $S_\alpha\uparrow$ | $E_\phi\uparrow$ | $F_\beta^\omega\uparrow$ | M↓ | |
| **CNN-based Methods** | | | | | | | | | | | | | |
| SINetV2(Fan et al., 2022) | 0.820 | 0.882 | 0.743 | 0.070 | 0.815 | 0.887 | 0.680 | 0.037 | 0.847 | 0.914 | 0.770 | 0.048 | - |
| SegMaR(Jia et al., 2022) | 0.815 | 0.884 | 0.753 | 0.071 | 0.833 | 0.906 | 0.724 | 0.033 | 0.841 | 0.907 | 0.781 | 0.046 | - |
| ZoomNet(Pang et al., 2022) | 0.820 | 0.892 | 0.752 | 0.066 | 0.838 | 0.911 | 0.729 | 0.029 | 0.853 | 0.912 | 0.784 | 0.043 | - |
| BSA-Net(Zhu et al., 2022) | 0.796 | 0.851 | 0.717 | 0.079 | 0.818 | 0.891 | 0.699 | 0.034 | 0.842 | 0.907 | 0.771 | 0.048 | - |
| BGNet(Sun et al., 2022) | 0.812 | 0.870 | 0.749 | 0.073 | 0.831 | 0.901 | 0.722 | 0.033 | 0.851 | 0.907 | 0.788 | 0.044 | - |
| DGNet(Ji et al., 2023) | 0.839 | 0.901 | 0.769 | 0.057 | 0.822 | 0.896 | 0.693 | 0.033 | 0.857 | 0.911 | 0.784 | 0.042 | - |
| FEDER(He et al., 2023) | 0.807 | 0.873 | 0.785 | 0.069 | 0.823 | 0.900 | 0.740 | 0.032 | 0.846 | 0.905 | 0.817 | 0.045 | - |
| CamoFormer-R(Yin et al., 2024) | 0.817 | 0.885 | 0.752 | 0.067 | 0.838 | 0.930 | 0.724 | 0.029 | 0.855 | 0.914 | 0.788 | 0.042 | - |
| ZoomNeXt-R(Pang et al., 2024) | 0.833 | 0.891 | 0.774 | 0.065 | 0.861 | 0.925 | 0.768 | 0.026 | 0.874 | 0.928 | 0.816 | 0.037 | - |
| **Transformer-based Methods** | | | | | | | | | | | | | |
| OSFormer(Pei et al., 2022) | 0.799 | 0.858 | 0.767 | 0.073 | 0.811 | 0.881 | 0.701 | 0.034 | 0.832 | 0.891 | 0.790 | 0.049 | - |
| MSCAF-Net(Liu et al., 2023) | 0.873 | 0.929 | 0.828 | 0.046 | 0.865 | 0.927 | 0.775 | 0.024 | 0.887 | 0.935 | 0.839 | 0.032 | - |
| HitNet(Hu et al., 2023) | 0.844 | 0.902 | 0.801 | 0.057 | 0.868 | 0.932 | 0.798 | 0.024 | 0.870 | 0.921 | 0.825 | 0.039 | - |
| FSPNet(Huang et al., 2023) | 0.856 | 0.928 | 0.799 | 0.050 | 0.851 | 0.930 | 0.735 | 0.026 | 0.879 | 0.937 | 0.816 | 0.035 | - |
| SARNet(Xing et al., 2023) | 0.874 | 0.929 | 0.844 | 0.046 | *0.885* | 0.941 | 0.820 | 0.021 | 0.889 | 0.934 | 0.851 | 0.032 | - |
| UEDG(Lyu et al., 2023) | 0.868 | 0.922 | 0.819 | 0.048 | 0.858 | 0.924 | 0.766 | 0.025 | 0.881 | 0.928 | 0.829 | 0.035 | - |
| CamoFormer-P(Yin et al., 2024) | 0.872 | 0.938 | 0.831 | 0.046 | 0.869 | 0.939 | 0.786 | 0.023 | 0.892 | *0.946* | 0.847 | 0.030 | - |
| ZoomNeXt-P(Pang et al., 2024) | 0.889 | 0.945 | *0.857* | *0.041* | **0.898** | **0.956** | 0.827 | 0.018 | 0.903 | 0.951 | 0.863 | 0.028 | - |
| **Diffusion-based Methods** | | | | | | | | | | | | | |
| diffCOD(Chen et al., 2023b) | 0.795 | 0.852 | 0.704 | 0.082 | 0.812 | 0.892 | 0.684 | 0.036 | 0.837 | 0.891 | 0.761 | 0.051 | 1000 |
| CamoDiffusion(Chen et al., 2024) | 0.871 | 0.940 | 0.849 | 0.043 | 0.868 | 0.940 | 0.803 | 0.021 | 0.887 | 0.941 | 0.857 | 0.029 | 10 |
| UGDNet(Yang et al., 2025) | *0.888* | *0.942* | 0.865 | 0.038 | *0.885* | *0.947* | *0.822* | *0.019* | 0.895 | 0.943 | *0.862* | 0.028 | 10 |
| CODiff (Ours) | **0.894** | **0.947** | **0.881** | **0.033** | 0.896 | 0.953 | **0.852** | **0.014** | **0.911** | **0.955** | **0.893** | **0.020** | **1** |

# 5. Experiment

## 5.1. Experimental Settings

**Datasets and Evaluation Metrics.** We evaluate our method on three widely used benchmarks for camouflaged object detection: CAMO (250 test images) (Le et al., 2019), COD10K (2,026 test images) (Fan et al., 2020), and NC4K (4,121 test images) (Lv et al., 2021). For quantitative comparison, we follow the previous work and adopt the following metrics: structure measure $S_\alpha$ (Fan et al., 2017), mean E-measure $E_\phi$ (Fan et al., 2018), weighted F-measure $F_\beta^\omega$ (Margolin et al., 2014), and mean absolute error M (Perazzi et al., 2012).

**Implementation Details.** We adopt the pretrained DINOv2-ViT-B14 model (Oquab et al., 2024) as a frozen vision backbone, and input resolution is $448 \times 448$. We use the AdamW optimizer with $betas = (0.9, 0.95)$. The learning rate is initialized to $1 \times 10^{-4}$ and follows a cosine annealing schedule. The entire model is trained until convergence with a batch size of 32 on four NVIDIA L40S GPUs. All experiments are conducted using the same random seed. More implementation details are provided in Appendix C.

## 5.2. Main Results

**Quantitative Analysis.** We compare our proposed method with 20 competing COD methods based on CNNs (Fan et al., 2022; Jia et al., 2022; Pang et al., 2022; Sun et al., 2022; Zhu et al., 2022; He et al., 2023; Ji et al., 2023; Pang et al., 2024; Yin et al., 2024), transformers (Pei et al., 2022; Hu et al., 2023; Huang et al., 2023; Liu et al., 2023; Xing et al., 2023; Lin et al., 2024; Pang et al., 2024; Yin et al., 2024) and diffusion models (Chen et al., 2023b; 2024; Yang et al., 2025) as shown in Tab. 1. The quantitative results demonstrate that our approach outperforms existing COD methods with NFE = 1. Specifically, our model reduces the MAE by 21.3% on average and improves $F_\beta$ by 2.8% on average compared to the second-best method.

**Qualitative Analysis.** We provide visual comparisons of COD masks predicted by our method and other representative methods, including CNN-based, transformer-based and diffusion-based models under several challenging scenarios. As shown in Fig. 3, the visual results demonstrate that our method achieves higher accuracy and produces clearer object boundaries than existing approaches. More visual results are provided in Appendix E.

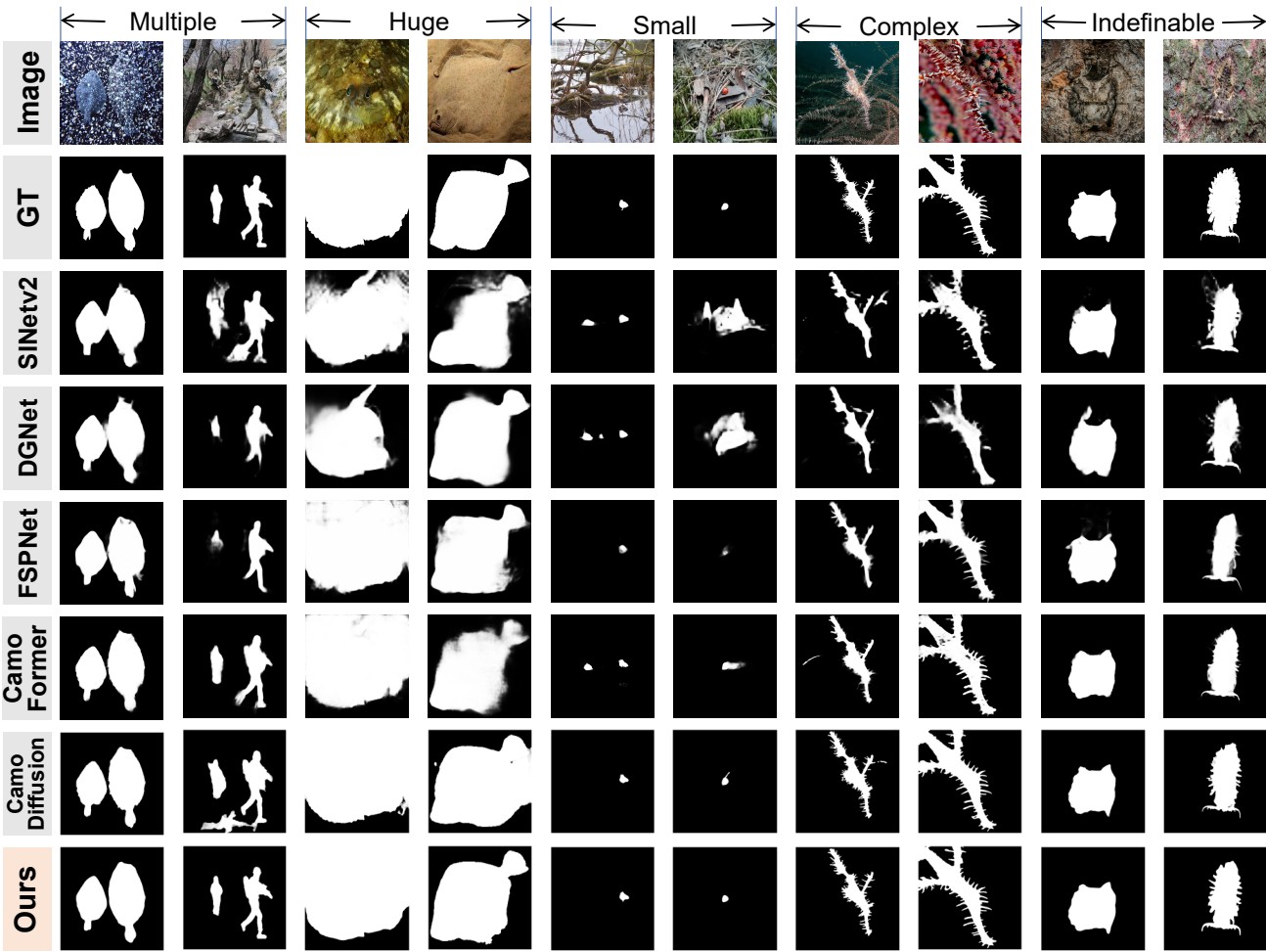

*Figure 3.* Visual comparisons of our method with other existing methods. Under various challenging scenarios, including multiple objects, huge targets, small targets, complex shapes, and indefinable boundaries, our method produces more accurate segmentation boundaries.

## 5.3. Ablation Study

In this subsection, we perform ablation experiments to validate the effectiveness of the proposed components, including the GSG mechanism, the HCI block, and $\mathcal{L}_{RA}$. Specifically, the ablation experiment is conducted on two representative datasets, COD10K and NC4K.

*Table 2.* Ablation study on GSG mechanism, including the global semantic token `<cls>` and the timestep embedding $t$.

| (`<cls>`, t) | COD10K | | | | NC4K | | | |
|---|---|---|---|---|---|---|---|---|
| | $S_\alpha\uparrow$ | $E_\phi\uparrow$ | $F_\beta^\omega\uparrow$ | $M\downarrow$ | $S_\alpha\uparrow$ | $E_\phi\uparrow$ | $F_\beta^\omega\uparrow$ | $M\downarrow$ |
| $(\times,\times)$ | 0.889 | 0.942 | 0.838 | 0.015 | 0.905 | 0.945 | 0.882 | 0.021 |
| $(\checkmark,\times)$ | 0.892 | 0.946 | 0.840 | 0.015 | 0.908 | 0.948 | 0.885 | 0.021 |
| $(\times,\checkmark)$ | 0.893 | 0.951 | 0.845 | 0.015 | 0.911 | 0.956 | 0.891 | 0.020 |
| $(\checkmark,\checkmark)$ | **0.896** | **0.953** | **0.852** | **0.014** | **0.911** | **0.955** | **0.893** | **0.020** |

**Guidance Ablation**. To validate the contribution of the proposed GSG mechanism, we conduct an ablation study

on the timestep $t$ and the `<cls>` token. Tab. 2 shows that removing either component leads to a decrease in COD performance, indicating that both the timestep $t$ and the global `<cls>` token play a crucial role in achieving accurate predictions. Moreover, when no guidance is applied, the COD performance reaches the lowest level.

*Table 3.* Ablation study on multi-level DINOv2 features in HCI.

| Layers | COD10K | | | | NC4K | | | |
|---|---|---|---|---|---|---|---|---|
| | $S_\alpha\uparrow$ | $E_\phi\uparrow$ | $F_\beta^\omega\uparrow$ | $M\downarrow$ | $S_\alpha\uparrow$ | $E_\phi\uparrow$ | $F_\beta^\omega\uparrow$ | $M\downarrow$ |
| [12] | 0.871 | 0.944 | 0.803 | 0.020 | 0.889 | 0.950 | 0.864 | 0.027 |
| [9,12] | 0.886 | 0.950 | 0.834 | 0.016 | 0.905 | 0.954 | 0.885 | 0.022 |
| [6,9,12] | 0.894 | 0.952 | 0.848 | 0.015 | 0.911 | 0.954 | 0.892 | 0.020 |
| [3,6,9,12] | **0.896** | **0.953** | **0.852** | **0.014** | **0.911** | **0.955** | **0.893** | **0.020** |

**Effectiveness of HCI**. To investigate the contribution of multi-level semantic features from DINOv2 in our HCI blocks, we perform ablation studies by introducing different numbers of DINOv2 features. To obtain multi-level

features, we select features from layers [3, 6, 9, 12] as representative candidates for evaluation. As shown in Tab. 3, using the final output (12th) layer provides limited improvement, as it fails to exploit multi-scale representations and locate camouflaged objects with complex shapes. In conclusion, further incorporating features from the [3,6,9,12] layers achieves the best performance, demonstrating that our model is capable of capturing strong representations for COD via multi-level condition features.

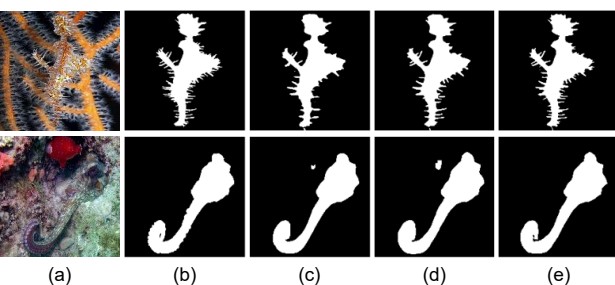

*Figure 4.* Visual comparisons of using different layers of DINOv2. a). Input condition image. b). GT mask. c) Using [9, 12] layers. d) Using [6, 9, 12] layers. e) Using [3, 6, 9, 12] layers. Visual results demonstrate that our choice for CODiff generates the most accurate predictions.

**Hyper-parameter Ablation**. We examine the effect of the regularization coefficient $\lambda$ in $\mathcal{L}_{total}$ with different values from 0 to 1. As shown in Tab. 4, the COD performance is saturated after $\lambda = 0.5$, and we apply $\lambda = 0.75$ to the other experiments.

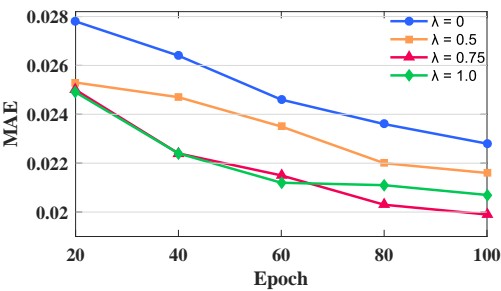

*Figure 5.* Hyperparameter sensitivity analysis. Convergence comparison with different $\lambda$. $\mathcal{L}_{RA}$ effectively accelerates convergence speed with the best $\lambda=0.75$.

*Table 4.* Ablation study on the hyper-parameter $\lambda$ of $\mathcal{L}_{RA}$.

| $\lambda$ | COD10K | | | | NC4K | | | |
|---|---|---|---|---|---|---|---|---|
| | $S_\alpha\uparrow$ | $E_\phi\uparrow$ | $F_\beta^\omega\uparrow$ | $M\downarrow$ | $S_\alpha\uparrow$ | $E_\phi\uparrow$ | $F_\beta^\omega\uparrow$ | $M\downarrow$ |
| w/o | 0.889 | 0.944 | 0.836 | 0.017 | 0.901 | 0.952 | 0.878 | 0.023 |
| 0.5 | 0.890 | 0.945 | 0.841 | 0.017 | 0.903 | 0.953 | 0.881 | 0.022 |
| 0.75 | **0.896** | **0.953** | **0.852** | **0.014** | **0.911** | **0.955** | **0.893** | **0.020** |
| 1.0 | 0.890 | 0.948 | 0.844 | 0.015 | 0.907 | 0.954 | 0.887 | 0.021 |

**Vision Backbone Ablation**. To verify that the superior performance of our model stems from a novel framework rather than from the pretrained vision backbone, we conduct the ablation experiment on different vision backbones as Tab. 5.

*Table 5.* Ablation study on different vision backbones on the COD10K dataset. F/T denotes whether the backbone is frozen (F) or trainable (T). CamoD denotes a previous diffusion-based method CamoDiffusion (Chen et al., 2024) with PVTv2 (Wang et al., 2022) as the condition encoder.

| Method | Backbone | $S_\alpha\uparrow$ | $E_\phi\uparrow$ | $F_\beta^\omega\uparrow$ | $M\downarrow$ |
|---|---|---|---|---|---|
| CamoD | PVTv2 (T) | 0.880 | 0.943 | 0.815 | 0.020 |
| CamoD | DINOv2 (F) | 0.870 | 0.930 | 0.799 | 0.021 |
| Ours | DINOv2 (F) | **0.896** | **0.953** | **0.852** | **0.014** |

**Semantic Gap**. We compare the representation gap at different timesteps to demonstrate $\mathcal{L}_{RA}$ has an effective impact on bridging the representation gap between the vision model and the diffusion model, as Fig. 6 illustrated. The CKNNA values demonstrate that our method has effectively aligned the intermediate feature space of the diffusion model with that of the condition vision model.

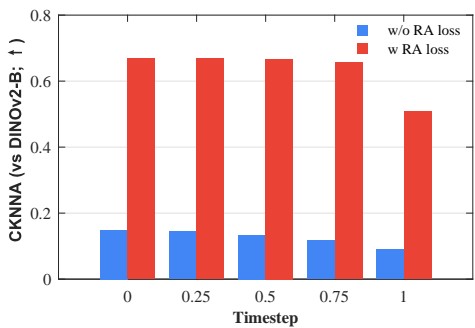

*Figure 6.* Representation gap across different timesteps. We plot the maximum CKNNA (Huh et al., 2024) scores at different timesteps, comparing the vanilla diffusion model without $\mathcal{L}_{RA}$ and the same model trained using $\mathcal{L}_{RA}$. $\mathcal{L}_{RA}$ consistently reduces the representation gap across different noise levels.

Moreover, we conduct a linear probing experiment (see Appendix B.1). Both the CKNNA scores and linear probing results indicate that our method effectively reduces the semantic gap between the diffusion model and the condition encoder, leading to improved representations. As a result, our model achieves better COD performance with faster convergence.

**Convergence and Sampling Analysis**. Regarding sampling efficiency, our model surpasses existing diffusion-based approaches as shown in the last column of Tab. 1 and Tab. 6, primarily benefiting from the formulated sample framework. More discussions are provided in Appendix D.

*Table 6.* Sampling comparison on COD10K dataset.

| Method | Sampling Time (s/img) | GFlops |
|---|---|---|
| CamoD | 0.18 | 506.65 |
| Ours | **0.07** | **254.41** |

## 6. Conclusion

In this paper, we revisit COD from a generative perspective and propose CODiff, a novel one-step diffusion-based framework that eliminates the reliance on iterative sampling. Motivated by our theoretical analysis on the feasibility of one-step generation in COD, we designed a customized architecture that effectively exploits conditional information and temporal guidance for accurate mask prediction in a single reverse step. Additionally, to facilitate efficient representation learning and improve convergence, we proposed a simple yet effective $\mathcal{L}_{RA}$. Extensive experiments on multiple benchmark datasets demonstrate that CODiff achieves competitive performance with only a single sampling step. We believe that this work provides new insights into bridging generative diffusion modeling and discriminative prediction tasks, and may inspire future research on efficient diffusion-based frameworks for other vision tasks beyond COD.

## Impact Statement

This paper presents work whose goal is to advance the field of Machine Learning. There are many potential societal consequences of our work, none which we feel must be specifically highlighted here.

## Acknowledgements

This work was supported by the National Natural Science Foundation of China under Grant Nos. 92367205 and U24A20258, and by the Natural Science Foundation of Zhejiang Province under Grant LR26F030002.

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

# A. Derivation of the Definition of $\mathbf{F}_{\boldsymbol{\theta}}(\mathbf{x}_t, t)$

According to Eq. 4, to solve the PF ODE is equivalent to calculating the following integral:

$$\int_T^0 \frac{d\mathbf{x}_t}{dt} dt = \int_T^0 -\frac{1}{2}\beta(t)\left[\mathbf{x}_t - \nabla_{x_t}\log q_t(\mathbf{x}_t)\right]dt \iff \mathbf{x}_0 = \mathbf{x}_T - \int_T^0 \frac{1}{2}\beta(t)\left[\mathbf{x}_t - \nabla_{x_t}\log q_t(\mathbf{x}_t)\right]dt \tag{13}$$

Thus we have:

$$\mathbf{x}_0 - \mathbf{x}_T = -\int_T^0 \frac{1}{2}\beta(t)\left[\mathbf{x}_t - \nabla_{x_t}\log q_t(\mathbf{x}_t)\right]dt \tag{14}$$

where $\mathbf{x}_T$ is initialized from a normal distribution $\mathcal{N}(0, \mathrm{I})$. We introduce $\mathbf{F}(\mathbf{x}_t, t, \mathbf{I})$ to represent the right-hand side of this equation and we have:

$$\mathbf{x}_0 - \mathbf{x}_t = -\mathbf{F}(\mathbf{x}_t, t, \mathbf{I}) \implies \mathbf{x}_0 = \mathbf{x}_t - \mathbf{F}(\mathbf{x}_t, t, \mathbf{I}) \tag{15}$$

In this work, we define a neural network-parameterized function $\mathbf{F}_{\boldsymbol{\theta}}(\mathbf{x}_t, t, \mathbf{I})$. Thus, our COD task turns to predict

$$\hat{\mathbf{x}}_0 = \mathbf{x}_t - \mathbf{F}_{\boldsymbol{\theta}}(\mathbf{x}_t, t, \mathbf{I}) \tag{16}$$

Under the above condition, obtaining accurate camouflaged object masks requires the model $\mathbf{F}_{\boldsymbol{\theta}}(\mathbf{x}_t, t, \mathrm{I})$ to be sufficiently close to $\mathbf{F}(\mathbf{x}_t, t, \mathrm{I})$.

# B. Evaluation of Representation Gap

### B.1. Definition of CKNNA

In this paper, we utilize the metric of Centered Kernel Nearest-Neighbor Alignment (CKNNA) to evaluate the representation gap between the diffusion model and the condition encoder (e.g. DINOv2 in this work), which is a relaxed version of the popular Centered Kernel Alignment (CKA) (Kornblith et al., 2019) that mitigates the strict definition. We generally follow the notations in the original paper to explain the conception CKNNA (Huh et al., 2024). First, CKA have measured global similarities of the models by considering all possible data pairs:

$$\mathrm{CKA}(\mathbf{K}, \mathbf{L}) = \frac{\mathrm{HSIC}(\mathbf{K}, \mathbf{L})}{\sqrt{\mathrm{HSIC}(\mathbf{K}, \mathbf{K})\mathrm{HSIC}(\mathbf{L}, \mathbf{L})}} \tag{17}$$

where $\mathbf{K}$ and $\mathbf{L}$ are two kernel matrices computed from the dataset using two different networks. Specifically, it is defined as $\mathbf{K}_{ij} = \kappa(\phi_i, \phi_j)$ and $\mathbf{L}_{ij} = \kappa(\psi_i, \psi_j)$ where $\phi_i, \phi_j$ and $\psi_i, \psi_j$ are representations computed from each network at the corresponding data $x_i, x_j$ (respectively). By letting $\kappa$ as a inner product kernel, HSIC is defined as

$$\mathrm{HSIC}(\mathbf{K}, \mathbf{L}) = \frac{1}{(n-1)^2}\left(\sum_i \sum_j \left(\langle\phi_i, \phi_j\rangle - \mathbb{E}_l[\langle\phi_i, \phi_l\rangle]\right)\left(\langle\psi_i, \psi_j\rangle - \mathbb{E}_l[\langle\psi_i, \psi_l\rangle]\right)\right) \tag{18}$$

CKNNA considers a relaxed version of Eq. (17) by replacing $\mathrm{HSIC}(\mathbf{K}, \mathbf{L})$ into $\mathrm{Align}(\mathbf{K}, \mathbf{L})$, where $\mathrm{Align}(\mathbf{K}, \mathbf{L})$ computes Eq. (18) only using a $k$-nearest neighborhood embedding in the datasets:

$$\mathrm{Align}(\mathbf{K}, \mathbf{L}) = \frac{1}{(n-1)^2}\left(\sum_i \sum_j \alpha(i, j)\left(\langle\phi_i, \phi_j\rangle - \mathbb{E}_l[\langle\phi_i, \phi_l\rangle]\right)\left(\langle\psi_i, \psi_j\rangle - \mathbb{E}_l[\langle\psi_i, \psi_l\rangle]\right)\right) \tag{19}$$

where $\alpha(i, j)$ is defined as

$$\alpha(i, j; k) = \mathbb{1}[i \neq j \text{ and } \phi_j \in \mathrm{knn}(\phi_i; k) \text{ and } \psi_j \in \mathrm{knn}(\psi_i; k)] \tag{20}$$

Hence, this term only considers $k$-nearest neighbors at each $i$.

In this paper, we use all images of two datasets, CAMO and COD10K to evaluate the CKNNA score with $k = 10$.

### B.2. Results of Linear Probing

To evaluate the semantic gap, we conduct linear probing experiments to demonstrate that our representation alignment loss $\mathcal{L}_{RA}$ can effectively enhance the learning of intermediate features to obtain more semantics.

*Table 7.* Linear probing results on COD10K and NC4K datasets.

| $\mathcal{L}_{RA}$ | COD10K | | | | NC4K | | | |
|---|---|---|---|---|---|---|---|---|
| | $S_\alpha\uparrow$ | $E_\phi\uparrow$ | $F_\beta^\omega\uparrow$ | $M\downarrow$ | $S_\alpha\uparrow$ | $E_\phi\uparrow$ | $F_\beta^\omega\uparrow$ | $M\downarrow$ |
| w/o | 0.842 | 0.923 | 0.762 | 0.023 | 0.873 | 0.937 | 0.838 | 0.030 |
| w | **0.880** | **0.936** | **0.825** | **0.017** | **0.893** | **0.939** | **0.865** | **0.025** |

## C. Further Implementation Details.

We use AdamW (Loshchilov & Hutter, 2019) with $(\beta_1, \beta_2) = (0.9, 0.95)$ and weight decay = 0.05. We adopt a cosine-annealing scheduler with an initial learning rate of 0.0001. To accelerate training, we use mixed-precision (FP16) with gradient clipping. We provide a detailed hyperparameter setup in Tab 8.

*Table 8.* Hyperparameter setup.

| | |
|---|---|
| Input dim. | $448 \times 448 \times 3$ |
| Num. layers | [2, 4, 6, 8] |
| Hidden dim. | [32, 64, 256, 768] |
| Num. heads | [4, 8, 16, 32] |
| Timestep dim. | 256 |
| Selected layers | [3, 6, 9, 12] |

## D. Discussions

**Revisiting Diffusion for COD.** Our work identifies a critical distinction often overlooked in previous diffusion-based COD methods: unlike natural image synthesis, which models a high-dimensional, multi-modal distribution to ensure diversity, COD maps complex visual features to low-entropy binary masks. Therefore, the inherent characteristic of the target distribution suggests the feasibility of one-step generation for COD, providing a highly efficient alternative to traditional iterative refinement.

**Limitations and Future Directions.** While the one-step formulation offers a clear computational advantage over multi-step diffusion sampling, the pursuit of more efficient architectures remains an open direction. Future work will explore lightweight decoder designs and compression techniques to further reduce computational overhead.

# E. Additional Prediction Samples

In this section, we provide additional qualitative examples from our model across different datasets.

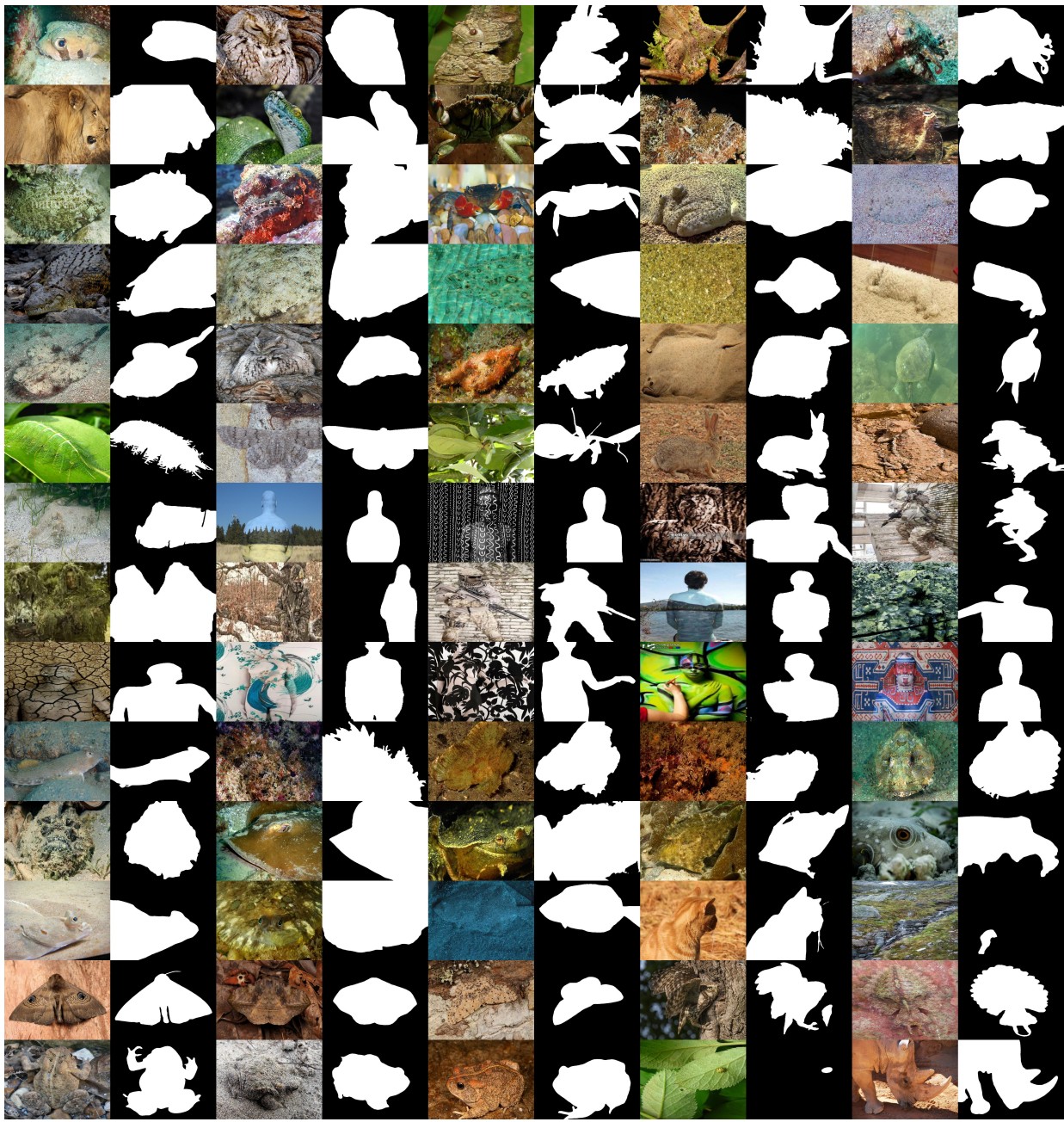

*Figure 7.* More one-step samples from CODiff model

