# OpenReview forum: "CODiff: One-Step Diffusion Model for Camouflaged Object Detection"
_ICML.cc/2026/Conference — ICML 2026 regular_

### Official Review · Reviewer_j2H2 · 2026-02-24

**Soundness:** 3
**Presentation:** 4
**Significance:** 3
**Originality:** 3
**Overall Recommendation:** 4
**Confidence:** 3

**Summary:**

CODiff is a one step model that makes object detection with a one step diffusion method on the task of COD. The author states that they get SOTA performance across multiple datasets.

The model uses a vision backbone with multilevel image features as help to generate the segmentation mask in the one step diffusion process. One step is enough when the model should be used for object detection and since the real image is inserted into the large backbone this seems possible.

**Compliance With Llm Reviewing Policy:**

Affirmed.

**Final Justification:**

The paper seems solid, it can be accepted from my point of view. I have changed the accept a bit down to my
Choice of
3: You are fairly confident in your assessment. It is possible that you did not understand some parts of the submission or that you are unfamiliar with some pieces of related work. Math/other details were not carefully checked.

**Key Questions For Authors:**

Q1: Have you evaluated CODiff with alternative vision backbones beyond DINOv2 and PVTv2? For example, DINOv3 or other recent vision foundation models could provide even stronger features. Results with newer backbones would help clarify how much of the performance gain comes from the proposed framework versus the backbone. This would increase my confidence in the contribution. DINOv3, for instance, has shown strong performance in efficient segmentation with very few examples, though I am unsure how it performs specifically in COD. Given the rapid progress in vision foundation models, it is important to understand whether the proposed framework remains beneficial when paired with increasingly powerful backbones, or whether a simpler decoder on top of a stronger backbone could achieve comparable results.

Q2: How does a simple baseline of DINOv2 features with a lightweight segmentation decoder (without the diffusion formulation) perform on the same benchmarks? If the gap is small, it would raise questions about the necessity of the diffusion framework. A strong gap would strengthen the paper.

Q3: Do the reported GFLOPs and inference times in Table 6 include the cost of the frozen DINOv2 backbone? If not, what is the total computational cost including feature extraction? This is important for a fair comparison with methods using lighter backbones.

**Limitations:**

yes

**Strengths And Weaknesses:**

Strengths:

S1: The use of a strong frozen vision backbone (DINOv2) combined with a dedicated one-step diffusion architecture is an effective approach for reducing the number of sampling steps while maintaining accuracy.

S2: The paper is very well written and easy to read. The figures are appropriately sized and self-explanatory, and the tables are clear and make it easy to compare methods.

S3: Comprehensive comparison against 20+ methods across CNN-based, Transformer-based, and diffusion-based approaches. CODiff achieves the best or second-best results across nearly all metrics.

S4: Overall, this is a very strong paper.

Weaknesses:

W1: The ablation in Table 5 suggests that the choice of vision backbone has a significant impact on performance. It would strengthen the paper to evaluate with additional or newer backbones to better disentangle the contribution of the proposed framework from the backbone itself. If a stronger backbone already closes most of the gap, the added value of the one-step diffusion framework becomes less clear.

W2: The paper does not compare against simply using DINOv2 features with a lightweight segmentation head (e.g., a simple decoder) as a baseline. This would help clarify how much the diffusion formulation itself contributes versus the strong features from DINOv2.

W3: The reported computational cost (Table 6) does not clearly state whether the GFLOPs and inference time include the frozen DINOv2 backbone or only the diffusion model. Since the method relies on extracting multi-level features from DINOv2 at inference time, the full computational overhead should be reported for a fair comparison with methods that use lighter backbones.

---

> ### Author Rebuttal · Authors · 2026-03-31
>
> We sincerely thank the reviewer for thoughtful and constructive feedback.
>
> **Rebuttal for Key Questions.**
>
> **Q1.**
> We conducted additional experiments by replacing the backbone with both a weaker model (DINOv1) and a more recent and stronger model (DINOv3).
>
> Across all settings, our framework consistently brings substantial improvements over a lightweight decoder built on the same backbone as shown in the tables below, specifically:
>
> - With DINOv1, our method improves performance by a large margin (e.g., on COD10K, $S_{\alpha}$ from 0.741 to 0.802, $F_{\beta}^{\omega}$ from 0.589 to 0.691, and MAE from 0.051 to 0.033).
> - With DINOv3, despite the already strong baseline, our full model still yields significant gains (e.g., on COD10K, $S_{\alpha}$ from 0.831 to 0.916, $F_{\beta}^{\omega}$ from 0.759 to 0.871, and MAE from 0.023 to 0.012).
>
> These results indicate that the performance improvements are not solely due to the backbone, but largely stem from the proposed framework, which remains effective across different backbone capacities.
>
> As expected, DINOv1 performs noticeably weaker than DINOv3, likely due to its more limited pre-training data and less advanced training strategy, resulting in weaker generalization as a frozen feature extractor. In contrast, newer vision foundation models provide stronger and more transferable representations. However, we did not adopt DINOv3 as our primary backbone for practical reasons: its weights currently require an official application, which may hinder accessibility for future users of our method. Our work demonstrates strong performance with readily available backbones, highlighting the practical applicability of our framework.
>
> Importantly, even when paired with a strong backbone, our method continues to provide substantial and non-trivial gains, demonstrating that it is complementary rather than redundant. Simply adopting a stronger backbone with a lightweight decoder is insufficient to match the performance of our full framework.
>
> In summary, these results strongly support that our method is consistently effective across a wide spectrum of representation qualities, providing compelling evidence that the observed performance gains primarily arise from the proposed framework, and that its benefits persist—even as backbone architectures continue to advance.
>
> |Setting|Dataset|$S_{\alpha}\uparrow$|$F_{\beta}^{\omega}\uparrow$|$E_{\phi}\uparrow$|$M\downarrow$|
> |:---|:---:|:---:|:---:|:---:|:---:|
> |DINOv1+Lightweight Decoder|CAMO|.736|.658|.840|.095|
> |-|COD10K|.741|.589|.845|.051|
> |-|NC4K|.795|.717|.882|.058|
> |DINOv1+Our Method|CAMO|.783|.721|.846|.072|
> |-|COD10K|.802|.691|.873|.033|
> |-|NC4K|.840|.782|.897|.041|
>
> |Setting|Dataset|$S_{\alpha}\uparrow$|$F_{\beta}^{\omega}\uparrow$|$E_{\phi}\uparrow$|$M\downarrow$|
> |:---|:---:|:---:|:---:|:---:|:---:|
> |DINOv3+Lightweight Decoder|CAMO|.838|.800|.906|.049|
> |-|COD10K|.831|.759|.924|.023|
> |-|NC4K|.873|.831|.945|.028|
> |DINOv3+Our Method|CAMO|.913|.905|.949|.032|
> |-|COD10K|.916|.871|.964|.012|
> |-|NC4K|.922|.910|.962|.015|
>
> **Q2.**
> To investigate the contribution of our framework beyond backbone choice, we utilize DINOv2 features with a lightweight segmentation decoder without the diffusion formulation as a controlled baseline.
>
> As shown in the table below, our full model consistently and substantially outperforms this baseline across all benchmarks. On COD10K, $S_{\alpha}$​ improves from 0.824 to 0.896, $F_{\beta}^{\omega}$ from 0.737 to 0.852, and MAE decreases from 0.027 to 0.014, with similar gains on CAMO and NC4K.
>
> When considered with the results in **Q1**, a consistent pattern emerges: our framework delivers evident improvements regardless of backbone strength — from weaker to stronger backbones.
>
> These results jointly demonstrate the effectiveness and generality of CODiff: it achieves strong COD performance while providing stable, substantial gains across diverse backbones, strongly suggesting that the improvements arise from the proposed diffusion-based framework rather than strong backbones.
> |Setting|Dataset|$S_{\alpha}\uparrow$|$F_{\beta}^{\omega}\uparrow$|$E_{\phi}\uparrow$|$M\downarrow$|
> |:---|:---:|:---:|:---:|:---:|:---:|
> |DINOv2+LightweightDecoder|CAMO|.823|.784|.902|.060|
> |-|COD10K|.824|.737|.915|.027|
> |-|NC4K|.861|.822|.934|.033|
> |DINOv2+Our Method|CAMO|.894|.881|.947|.033|
> |-|COD10K|.896|.852|.953|.014|
> |-|NC4K|.911|.893|.955|.020|
>
> **Q3.**
> The GFLOPs and inference time reported in Table 6 both include the computational cost of the frozen DINOv2 backbone. This ensures a fair comparison with other methods, including those using lighter backbones.
>
> **Rebuttal for Weaknesses.** Since the listed weaknesses correspond to the key questions, we have provided responses in the **Rebuttal for Key Questions** section.

---

> > ### Author Rebuttal · Reviewer_j2H2 · 2026-04-03
> >
> > The paper can be accepted from my point of view.

---

> > > ### Author Response · Authors · 2026-04-06
> > >
> > > Dear Reviewer j2H2,
> > >
> > > Thank you for your constructive and insightful feedback, and for recognizing the contributions of CODiff. We sincerely appreciate the time and effort you devoted to reviewing our work. Your thoughtful comments have helped us further highlight the key merits of our method. We will incorporate the improvements into the final version.
> > >
> > > Best regards,
> > >
> > > The Authors

---

### Official Review · Reviewer_rjjg · 2026-03-12

**Soundness:** 3
**Presentation:** 3
**Significance:** 2
**Originality:** 2
**Overall Recommendation:** 4
**Confidence:** 4

**Summary:**

This paper addresses the computational inefficiency and unnecessary complexity of existing diffusion-based Camouflaged Object Detection (COD) methods, which typically rely on iterative multi-step sampling to refine segmentation masks. The authors rethink COD as a structured prediction task with low-complexity conditional distributions, proposing CODiff, a novel framework that enables accurate mask prediction in a single reverse step. The methodology integrates a Global Semantic Guidance (GSG) mechanism for temporal and semantic awareness, Hierarchical Condition Integration (HCI) blocks to leverage multi-scale features from a DINOv2 backbone, and a Representation Alignment Loss ($L_{RA}$) to bridge the gap between the frozen encoder and the diffusion model. Extensive benchmarks on CAMO, COD10K, and NC4K datasets demonstrate that CODiff achieves state-of-the-art performance, significantly reducing Mean Absolute Error (MAE) and improving inference speed by eliminating iterative refinement.

**Compliance With Llm Reviewing Policy:**

Affirmed.

**Final Justification:**

The authors addressed all my concerns and I am willing to upgrade my rating.

**Key Questions For Authors:**

1. How exactly does the frozen DINOv2 backbone (pre-trained on RGB images) process binary "clean masks" to generate the target features $F_4^{gt}$ for the $L_{RA}$ calculation?
2. Can the authors provide a deterministic regression baseline that uses the exact same DINOv2 backbone and HCI architecture (trained with BCE/IoU, without the diffusion process) to empirically isolate the performance gain brought specifically by the diffusion formulation?
3. Could the authors provide the GFLOPs and parameter counts for the different multi-level feature combinations shown in Table 3?
4. How is the "inherently low-complexity" property of COD masks mathematically leveraged in the convergence analysis? Currently, Theorem 4.3 seems to rely only on general Lipschitz continuity.

**Limitations:**

The authors discussed limitations in the Appendix but did not discuss potential negative societal impact of their work.

**Strengths And Weaknesses:**

Strengths:
1. Theoretical Innovation in Diffusion Efficiency: The authors provide a theoretical formulation using Probability Flow (PF) ODEs to justify the feasibility of one-step reverse mapping for COD, transitioning the task from a stochastic generation process to an efficient structured prediction.
2. Effective Global-Local Feature Fusion: The design of the GSG mechanism successfully incorporates global <cls> tokens and temporal embeddings to guide the denoising process, while the HCI blocks ensure fine-grained structural semantics for complex camouflaged boundaries.
3. Representation Alignment Strategy: The introduction of $L_{RA}$ is a principled way to align the intermediate feature space of the diffusion model with the pre-trained vision backbone, facilitating faster convergence and better feature utilization without fine-tuning the backbone.
4. Superior Performance-Efficiency Trade-off: The model achieves competitive or superior results compared to multi-step diffusion and transformer-based models while significantly reducing GFlops and sampling time (e.g., 0.07s/img vs. 0.18s/img).

Weaknesses:
1. Since DINOv2 is strictly pre-trained on natural RGB images, please clarify how the frozen DINOv2 backbone generates target features $F_4^{gt}$ from binary "clean masks" for the $L_{RA}$ loss.
2. To genuinely isolate the benefits of the diffusion generative formulation, the authors should include a direct ablation baseline: a deterministic regression model using the exact same frozen DINOv2 backbone and HCI architecture, trained merely with standard segmentation losses (BCE, IoU) without any diffusion process or timestep conditioning.
3. The ablation study for the HCI module (Table 3) only reports accuracy metrics when aggregating multi-level DINOv2 features. Since introducing multiple high-dimensional features inevitably increases memory and computational costs, please include GFLOPs and parameter counts in Table 3 to demonstrate that the performance gains do not simply come from higher computational overhead.
4. The Introduction claims that the "inherently low-complexity" of the COD mask distribution justifies one-step sampling. However, the convergence analysis in Section 4.2 (Assumption 4.2 and Theorem 4.3) relies entirely on Lipschitz continuity and boundedness, without mathematically formulating or leveraging the "low-complexity" property. Bridging this theoretical gap would significantly strengthen the paper's core claim.
5. In Eq. (12), the total loss incorporates $L_{wBCE}$ and $L_{wIOU}$. While the text states these are "weighted" losses, the weighting factor is neither formally defined nor is its calculation mechanism explained. Please explicitly formulate to ensure mathematical rigor or include corresponding references.

---

> ### Author Rebuttal · Authors · 2026-03-31
>
> We sincerely thank the reviewer for thoughtful and constructive feedback.
>
> **Q1.** We replicate the binary mask to three channels to match DINOv2's input format. Despite being frozen, DINOv2 still produces discriminative features owing to its strong self-supervised pretraining and generalization capability.
>
> **Q2.**
> | Method              | $S_{\alpha}\uparrow$ | $F_{\beta}^{\omega}\uparrow$ | $E_{\phi}\uparrow$ | $M\downarrow$ |
> |:-------------------|:--------------------:|:----------------------------:|:------------------:|:-------------:|
> | HCI (w/o Diffusion) | .862                 | .824                         | .917               | .049          |
> | HCI (Diffusion)     | .873                 | .846                         | .926               | .041          |
> | CODiff              | .894                 | .881                         | .947               | .033          |
>
> Introducing diffusion consistently improves all metrics, confirming its effectiveness. CODiff achieves further gains, indicating synergy with our proposed components.
>
> **Q3**
> | Layers       | $S_{\alpha} \uparrow$ | $F_{\beta}^{\omega} \uparrow$ | $E_{\phi} \uparrow$ | $M \downarrow$ | GFLOPs (Δparam) |
> |:-----------|:-------------------:|:----------------------------:|:-----------------:|:-------------:|:---------------:|
> | [12]        | .871| .944 | .803              | .020          | 224.13 (≈−20M) |
> | [9,12]      | .886 | .950 | .834              | .016          | 239.23 (≈−10M) |
> | [6,9,12]    | .894 | .952| .848              | .015          | 244.74 (≈−10M) |
> | [3,6,9,12]  | .896  | .953  | .852              | .014          | 254.41 (baseline) |
>
> We have added the GFLOPs and parameter changes for each configuration. Although the \[9,12\] and \[6,9,12\] configurations have similar GFLOPs and parameters, the latter still shows noticeable improvement across all metrics, indicating that the performance gain comes from more effective feature fusion rather than simply higher computational cost.
>
> **Q4.**
> We will refine the theoretical derivations in the future version by incorporating the following steps, thereby establishing a complete and rigorous reasoning chain.
>
> **Assumption 4.5\*.**
> Since Remark 4.4 in the main paper asserts that $\mathbf{x}_0$ is uniquely determined by $\mathbf{I}$, setting $\mathbf{x}^* (\mathbf{I}):=\mathbb{E}[\mathbf{x}\_0\mid\mathbf{I}]$ yields:
> $$\mathrm{Var}[\mathbf{x}_0\mid\mathbf{I}]=\mathbb{E}\left[\|\mathbf{x}_0-\mathbf{x}^*(\mathbf{I})\|^2\mid\mathbf{I}\right]\leq\epsilon^2$$
>
> where $\epsilon\approx 0$ reflects annotation noise rather than semantic ambiguity. Thus Assumption 4.5* is not an additional restriction, but a formal quantification of the deterministic nature already stated in Remark 4.4.
>
> **Corollary 4.6\* (One-Step Error Bound).**
> Under Assumptions 4.2, 4.5\* and Theorem 4.3:
> $$\mathbb{E} \left[d\left(\hat{\mathbf{x}}\_0,\,\mathbf{x}^*(\mathbf{I})\right)\right] \leq C\cdot\mathcal{L}_{\mathrm{total}}(\theta) + \epsilon$$
>
> **Proof.** Triangle inequality gives:
>
> $$d(\hat{\mathbf{x}}_0, \mathbf{x}^* ) \leq d(\hat{\mathbf{x}}_0, \mathbf{x}_0) + d(\mathbf{x}_0, \mathbf{x}^* )$$
>
> Taking expectations:
>
> - **Term (i):** By Theorem 4.3, $\mathbb{E}[d(\hat{\mathbf{x}}\_0,\mathbf{x}\_0)] \leq C\cdot\mathcal{L}\_{\mathrm{total}}(\theta)$.
>
> - **Term (ii):** By Jensen's inequality and Assumption 4.5\*:
> $$\mathbb{E}[d(\mathbf{x}_0,\mathbf{x}^* )]\leq \mathbb{E}[\|\mathbf{x}_0-\mathbf{x}^* \|]\leq\sqrt{\mathbb{E}[\|\mathbf{x}_0-\mathbf{x}^* \|^2]}\leq\epsilon$$
>
> Combining yields the result. $\square$
>
> ---
>
> **Justification for One-Step Inference.**
> Standard conditioning reduces variance monotonically:
> $$\mathrm{Var}[\mathbf{x}_0\mid\mathbf{x}_t,\mathbf{I}]\leq\mathrm{Var}[\mathbf{x}_0\mid\mathbf{I}]\leq\epsilon^2$$
>
> Since $\mathrm{Var}[\mathbf{x}_0\mid\mathbf{I}]$ is already $O(\epsilon^2)$ by Assumption 4.5*, additional denoising steps yield negligible variance reduction. Thus one-step inference suffices theoretically, distinguishing COD from generative tasks where high conditional variance necessitates iterative refinement.
>
> **W1,2,3,4**:  See **Q1,2,3,4**.
>
> **W5**
> Our weighted losses follow Eq. (12):
>
> $$\mathcal{L}\_{\omega\mathrm{BCE}} = \frac{\sum\_{i,j} \omega\_{i,j} \cdot \mathrm{BCE}(P\_{i,j}, M_{i,j})}{\sum\_{i,j} \omega\_{i,j}}, \quad \mathcal{L}\_{\omega\mathrm{IoU}} = 1 - \frac{\sum\_{i,j} \omega\_{i,j} P\_{i,j} M_{i,j} + 1}{\sum\_{i,j} \omega\_{i,j}(P\_{i,j} + M\_{i,j} - P\_{i,j} M\_{i,j}) + 1}$$
>
> where $\omega\_{i,j}$ is a spatially adaptive weight emphasizing boundary regions, computed as the discrepancy between the ground-truth mask and its locally smoothed counterpart. We adopt the same weighting scheme as the previous diffusion-based method[1], fixed across all experiments.
>
> \[1\] Chen, Z., Sun, K., and Lin, X. CamoDiffusion: Camouflaged object detection via conditional diffusion models. In Proceedings of the AAAI Conference on Artificial Intelligence, volume 38, pp. 1272–1280, 2024.

---

> > ### Author Rebuttal · Reviewer_rjjg · 2026-04-03
> >
> > The authors addressed all my concerns and I am willing to upgrade my rating.

---

> > > ### Author Response · Authors · 2026-04-06
> > >
> > > Dear Reviewer rjjg,
> > >
> > > Thank you for your constructive and insightful feedback, and for recognizing the contributions of CODiff. We sincerely appreciate the time and effort you devoted to reviewing our work. Your thoughtful comments have helped us further highlight the key merits of our method. We will incorporate the improvements into the final version.
> > >
> > > Best regards,
> > >
> > > The Authors

---

### Official Review · Reviewer_kncR · 2026-03-13

**Soundness:** 3
**Presentation:** 3
**Significance:** 3
**Originality:** 3
**Overall Recommendation:** 4
**Confidence:** 3

**Summary:**

This paper proposes CODiff, a one-step diffusion-based framework for camouflaged object detection. Unlike existing diffusion-based COD methods that require multiple sampling steps, the authors reformulate the diffusion process to enable one-step mask prediction while maintaining competitive performance. The paper provides a theoretical formulation showing the feasibility of one-step sampling using a probability flow formulation, and designs a dedicated architecture including a Global Semantic Guidance mechanism, Hierarchical Condition Integration blocks, and a representation alignment loss to improve feature consistency between the diffusion model and the condition backbone. Experiments on several COD benchmarks demonstrate competitive performance compared with existing methods.

**Compliance With Llm Reviewing Policy:**

Affirmed.

**Final Justification:**

The paper can be accepted from my point of view.

**Key Questions For Authors:**

1.	The paper argues that existing diffusion-based COD methods are slow due to multi-step sampling, and proposes one-step diffusion for efficiency. Could the authors provide more comprehensive runtime comparisons, including multi-step diffusion variants implemented under the same framework, as well as strong non-diffusion baselines, to better support the claim of improved efficiency?
	2.	The proposed method contains several components, including the one-step diffusion formulation, Global Semantic Guidance, Hierarchical Condition Integration, and the representation alignment loss. Could the authors provide a more progressive ablation study starting from a minimal baseline and adding each component step by step, so that the contribution of each module can be more clearly understood?
	3.	Although the proposed method reduces the number of diffusion steps, the overall model still includes a diffusion UNet, a large vision backbone, and multiple conditional modules.
Could the authors provide a more complete efficiency analysis, including overall runtime, and FLOPs, compared with strong CNN-based or transformer-based COD methods?
	4.	The experimental section mainly reports overall performance, but does not provide detailed analysis of failure cases or difficult scenarios. Could the authors include more qualitative or quantitative analysis on challenging cases, such as small objects, complex backgrounds, or weak camouflage, to better illustrate the strengths and limitations of the proposed method?

**Limitations:**

yes

**Strengths And Weaknesses:**

Strength
The paper is generally well organized with a logical motivation that follows from the limitations of existing diffusion-based COD methods. The authors present the problem in a structured way, starting from the inefficiency of multi-step diffusion sampling and leading to the proposed one-step formulation. The methodology section is relatively detailed, including both theoretical analysis and architectural design. The experimental section covers multiple widely used COD benchmarks.

Weakness
1.The paper argues that existing diffusion-based COD methods are slow due to multi-step sampling, and proposes one-step diffusion for efficiency. However, the experiments do not provide sufficient comparisons of inference time with multi-step diffusion variants implemented under the same framework. In addition, the paper does not include speed comparisons with strong non-diffusion baselines.
2.The method introduces several key components, including one-step diffusion formulation, Global Semantic Guidance, Hierarchical Condition Integration, and the representation alignment loss. However, the ablation study does not clearly show a progressive analysis starting from a minimal baseline and gradually adding each component. Without such step-by-step ablation, it is difficult to understand which component contributes most to the final performance and whether the full complexity of the model is necessary.
3.The main contribution of the paper is the one-step diffusion formulation, but the experiments do not provide a direct comparison between one-step and multi-step sampling within the same framework.
4.Although the proposed method reduces the number of diffusion steps, the overall architecture still includes a diffusion UNet, a large vision backbone, and multiple conditional modules. It is unclear whether the proposed approach is actually more efficient than strong CNN-based or transformer-based COD methods when considering the full model complexity rather than only the sampling step.
5.	The experimental results mainly report overall performance but do not analyze scenarios where the proposed method fails or performs similarly to existing approaches. More detailed analysis on challenging cases, such as small objects, complex backgrounds, or weak camouflage, would help better understand the limitations of the proposed framework.

---

> ### Author Rebuttal · Authors · 2026-03-31
>
> We sincerely thank the reviewer for thoughtful and constructive feedback.
>
> **Rebuttal for Key Questions.**
>
> **Q1.**
> We report the inference time of representative non-diffusion COD methods as shown in the table below. Our method achieves competitive efficiency, approaching the strong non-diffusion models (CNN-based and Transformer-based) while delivering superior prediction performance.
>
> The main goal of this work is to explore whether a one-step diffusion paradigm can reduce computational overhead while maintaining strong predictive capability. Compared to conventional multi-step diffusion methods, our one-step design eliminates repeated sampling, achieving a better trade-off between efficiency and performance.
>
> We acknowledge, as discussed in Appendix D, that diffusion-based frameworks still have limitations in inference speed. We hope this work provides a promising direction for making diffusion models more practical in COD and inspires future research on improving efficiency while maintaining competitive COD performance.
> |Method|SamplingTime(s/image)|GFlops|
> |:---|:---:|:---:|
> |ZoomNeXt-ResNet|0.03|115.10|
> |ZoomNeXt-P|0.05|156.44|
> |FSPNet|0.04|283.48|
> |CODiff(ours)|0.07|254.41|
>
> **Q2.** We start from a basic diffusion framework and incrementally add each proposed component. Experiments on CAMO, COD10K, and NC4K show that all three components consistently improve baseline performance across multiple metrics. This demonstrates that each component positively contributes to the overall model and that improvements are consistent across multiple datasets.
> |Dataset|Method|$S_{\alpha}\uparrow$|$F_{\beta}^{\omega}\uparrow$|$E_{\phi}\uparrow$|$M\downarrow$|
> |:---:|:---:|:---:|:---:|:---:|:---:|
> |CAMO|-|.835|.806|.912|.053|
> |-|+HCI|.873|.846|.926|.041|
> |-|+GSG|.876|.852|.937|.038|
> |-|+LRA|.874|.854|.937|.040|
> |COD10K|-|.828|.741|.928|.026|
> |-|+HCI|.882|.826|.942|.017|
> |-|+GSG|.859|.789|.934|.021|
> |-|+LRA|.863|.798|.942|.020|
> |NC4K|-|.860|.824|.939|.034|
> |-|+HCI|.903|.879|.951|.022|
> |-|+GSG|.889|.859|.947|.026|
> |-|+LRA|.891|.864|.952|.025|
>
> **Q3.** See **Q1**.
>
>  **Q4.** In Figure 3 of the main paper and Appendix E, we have conducted qualitative analyses on various challenging scenarios, including multiple-object scenes, small objects, and camouflaged targets with complex shapes. The results demonstrate that the proposed method performs well on the vast majority of such challenging cases. We plan to include additional visual examples in the final version to provide a more comprehensive analysis.
>
> **Rebuttal for Weaknesses.**
> Since the listed weaknesses correspond to the key questions, we have provided responses in the **Rebuttal for Key Questions** section.

---

> > ### Author Rebuttal · Reviewer_kncR · 2026-04-01
> >
> > The author has addressed most of my concerns, but the content and experimental design still require further refinement.

---

> > > ### Author Response · Authors · 2026-04-03
> > >
> > > Thank you for your follow-up and for acknowledging our rebuttal.
> > >
> > > We realize that our explanations in the previous response were perhaps not detailed enough, which may have caused some confusion. Based on your latest comments, we understand that your remaining concerns might primarily revolve around the efficiency of our method and its performance in challenging scenarios. To provide a clearer and more comprehensive perspective, we have further elaborated on these aspects on top of our rebuttal and provided additional experiments results.
> > >
> > > **(1) About Efficiency**
> > >
> > > For COD tasks, true efficiency should be measured as achieving the best results with less computation, rather than simply
> > > improving speed at the cost of accuracy. To this end, we provide a comprehensive comparison including both non-diffusion and diffusion-based methods in the table below.
> > > | Method        | Sampling Time (s/image) | GFLOPs | $\mathbf{S_{\alpha} \uparrow}$ | $\mathbf{E_{\phi} \uparrow}$ | $\mathbf{F_{\beta}^{\omega} \uparrow}$ |    M↓     |
> > > | :------------ | :--------------: | :----: | :-----------: | :-----------: | :-----------: | :-------: |
> > > | ZoomNeXt-R |0.03 | 115.10 | 0.833 |0.891|  0.774  |   0.065   |
> > > | FSPNet        |0.04| 283.48 |0.856  |  0.928 | 0.799 |   0.050   |
> > > | ZoomNeXt-P    | 0.05 | 156.44 |_0.889_   |  _0.945_  |  _0.857_    |  _0.041_ |
> > > | CamoDiffusion | 0.18| 506.65 | 0.871|0.940  |  0.849  |   0.043   |
> > > | CODiff (Ours) | 0.07 | 254.41 | **0.894**  |  **0.947**    | **0.881**  | **0.033** |
> > >
> > > As shown, our method achieves **SOTA performance** across all metrics while maintaining competitive inference speed. Notably, compared to CamoDiffusion, the most representative multi-step diffusion baseline, our one-step design is **2.6× faster** while delivering consistently superior predictions.
> > >
> > > We acknowledge that diffusion-based frameworks still have limitations in inference speed compared to end-to-end CNN/Transformer-based methods. However, this is a fundamental and well-recognized limitation of the diffusion paradigm rather than a shortcoming specific to our method. It has motivated a growing line of research on one-step diffusion acceleration [1], and as discussed in our paper, remains an important direction for our future work.
> > >
> > > We further measure the inference time of our framework under different numbers of sampling steps. As shown in the table below, our single-step inference is **4.6× faster** than 5-step sampling and **8.9× faster** than 10-step sampling within the same framework, clearly demonstrating the efficiency advantage of our one-step formulation.
> > > |**NFEs**|**Sampling Time (s/img)**
> > > |:---:|:---:|
> > > |1|0.07|
> > > |5|0.32|
> > > |10|0.62|
> > >
> > > **(2) CODiff Performance on Challenging Scenarios**
> > >
> > > To demonstrate the practical effectiveness of our method in challenging scenarios, we provide a comprehensive set of qualitative and quantitative results, covering small objects, multiple objects, complex backgrounds, camouflaged objects with intricate shapes, occlusions, and weakly camouflaged objects. The corresponding visualizations are available at the following link [[anonymous link]](https://anonymous.4open.science/r/Anonymous_link). Although none of the methods, including ours, achieve ideal performance in some certain extreme scenarios, our approach still produces the most compelling results, whereas alternative methods often suffer from significant performance degradation.
> > >
> > > Furthermore, we conducted a quantitative assessment specifically on these difficult cases. The results as shown in the below table demonstrate that our method consistently maintains superior performance, highlighting the robustness and significant potential of diffusion-based frameworks for camouflaged object detection.
> > >
> > > |Method| $\mathbf{S_{\alpha} \uparrow}$ | $\mathbf{E_{\phi} \uparrow}$ | $\mathbf{F_{\beta}^{\omega} \uparrow}$ | M↓     |
> > > |:---|:---:|:---:|:---:|:---:|
> > > |FSPNet|*0.577*|*0.542*|0.258|*0.135*|
> > > |CamoFormer-P|0.511|0.467|0.200|0.195|
> > > |camodiffusion|0.549|0.523|*0.293*|0.194|
> > > |CODiff(Ours)|**0.886**|**0.943**|**0.822**|**0.020**|
> > >
> > > Thank you again for your constructive comments, which have helped us further highlight the advantages of our proposed method and emphasize our contributions. We will incorporate these improvements in the Appendix of the final version.
> > >
> > > [1] Geng, Z., Deng, M., Bai, X., Kolter, J. Z., & He, K. (2025). Mean flows for one-step generative modeling. *NeurIPS 2025*.

---

### Official Review · Reviewer_B246 · 2026-03-13

**Soundness:** 3
**Presentation:** 3
**Significance:** 3
**Originality:** 2
**Overall Recommendation:** 4
**Confidence:** 3

**Summary:**

This paper proposes a one-step diffusion-based model, 'CODiff', for Camouflaged Object Detection (COD). The key idea is to reformulate COD as a single-step reverse diffusion problem.

To achieve this, the authors designed 1) a Global Semantic Guidance (GSG) mechanism to guide the denoising process globally, 2) Hierarchical Condition Integration (HCI) blocks to provide fine-grained structural semantics, and 3) Representation Alignment Loss.

Experiments on three standard COD benchmarks show that the proposed CODiff method outperforms existing CNN-, Transformer-, and diffusion-based COD approaches on quantitative metrics.

**Compliance With Llm Reviewing Policy:**

Affirmed.

**Final Justification:**

My concerns have been addressed. The authors' responses and additional evidence support my original rating.

**Key Questions For Authors:**

1. The Global Semantic Guidance module is described as guiding the denoising process with global semantic information. Could the authors provide more intuition or visualisation to explain how this guidance influences the diffusion process and improves segmentation quality?

2. It would be helpful to provide a more controlled comparison to clarify where the performance gains actually come from. In particular, it is currently unclear how much of the improvement should be attributed to the one-step inference formulation itself, and how much comes from the stronger conditioning architecture, including GSG, HCI, and L_RA. For example, under the same backbone and training recipe, how does a simpler one-step baseline without GSG/HCI/L_RA perform?

**Limitations:**

Nil

**Strengths And Weaknesses:**

Strengths:

1. The empirical performance on standard COD benchmarks is strong, where CODiff achieves the best results across all reported metrics on CAMO, COD10K, and NC4K, demonstrating clear improvements over prior diffusion-based approaches.

2. The qualitative visualisations are convincing. The results shown in Figures 3 and 4 are generally consistent with the claims in the paper, particularly in cases involving thin structures and complex object boundaries, where the model demonstrates improved localisation ability.

3. The paper is clearly presented. The motivation and architecture diagrams make the pipeline easy to follow, component-level ablations are clearly organised, and the inclusion of training details and supplementary appendices helps improve reproducibility.

Weaknesses:

1. The experimental evaluation does not include the CHAMELEON dataset, which is widely adopted in prior COD studies as a standard benchmark. Including results on this dataset would improve the completeness of the experimental comparison and strengthen the overall credibility of the empirical evaluation.

2. There appear to be a few minor issues in the visualisations. In Figure 2, the images labeled Pred and GT appear to be identical, which may be an unintentional duplication. In Figure 3, within the “Multiple” example, the GT image appears to be mirrored, which needs to be corrected.

3. The examples shown in Figure 3 mainly highlight favourable cases where the proposed method performs well. However, the paper does not include examples illustrating more challenging scenarios, such as cases where the method fails, situations in which iterative diffusion approaches may produce better results than the proposed one-step prediction, or scenarios where the one-step prediction method has clear advantages over iterative diffusion methods. Providing both types of comparisons would help clarify under what conditions each paradigm is preferable.

4. The efficiency advantage of the proposed one-step design is also not fully demonstrated. Although the paper emphasises the reduction of diffusion sampling steps, it does not clearly report detailed runtime comparisons, inference latency, or memory consumption relative to existing multi-step diffusion COD models. It is therefore difficult to assess whether the one-step formulation leads to meaningful practical benefits.

5. Relatedly, the novelty at the component level appears somewhat limited. The proposed modules are reasonable and well-motivated, but each one is largely built on familiar design patterns: GSG uses global semantic and timestep information for feature modulation, HCI performs multi-scale cross-attention-based condition fusion, and L_RA introduces a cosine-similarity alignment regularizer. As a result, the paper’s main novelty seems to rely primarily on the overall one-step diffusion formulation, yet this aspect is not sufficiently isolated or analysed in either the theoretical or experimental sections.

6. The experimental comparison is missing several recent high-performing COD methods, such as FGSA-Net (Zhang et al., 2025), SCOUT (Yan et al., 2025b), RISE (Du et al., 2025b), EASE (Du et al., 2025a), and UCOD-DPL (Yan et al., 2025a):

[1] Zhang, S., Kong, D., Xing, Y., Lu, Y., Ran, L., Liang, G., ... & Zhang, Y. (2025). Frequency-guided spatial adaptation for camouflaged object detection. IEEE Transactions on Multimedia, 27, 72-83.
[2] Yan, W., Chen, L., Zhang, S., Zhang, Y., & Cao, L. (2025). SCOUT: Semi-supervised Camouflaged Object Detection by Utilizing Text and Adaptive Data Selection. arXiv preprint arXiv:2508.17843.
[3] Du, J., Wang, X., Hao, F., Yu, M., Chen, C., Wu, J., ... & Li, P. (2025). Beyond Single Images: Retrieval Self-Augmented Unsupervised Camouflaged Object Detection. In Proceedings of the IEEE/CVF International Conference on Computer Vision (pp. 22131-22142).
[4] Du, J., Hao, F., Yu, M., Kong, D., Wu, J., Wang, B., ... & Li, P. (2025). Shift the lens: Environment-aware unsupervised camouflaged object detection. In Proceedings of the Computer Vision and Pattern Recognition Conference (pp. 19271-19282).
[5] Yan, W., Chen, L., Kou, H., Zhang, S., Zhang, Y., & Cao, L. (2025). UCOD-DPL: Unsupervised Camouflaged Object Detection via Dynamic Pseudo-label Learning. In Proceedings of the Computer Vision and Pattern Recognition Conference (pp. 30365-30375).

---

> ### Author Rebuttal · Authors · 2026-03-31
>
> We sincerely thank the reviewer for thoughtful and constructive feedback.
>
> **Q1.**
> Intuitively, GSG integrates two complementary sources: (1) global semantics from the DINOv2 [CLS] token, providing high-level semantic guidance; and (2) noise-stage information from timestep $t$, indicating the denoising progress.
>
> GSG performs conditioned channel-wise modulation: it applies adaptive scaling and shifting to each channel of the features based on the conditional vector, enhancing target-aligned channels while suppressing background-dominated ones. This produces more discriminative representations, which is especially important for COD where local textures often resemble the background.
>
> Beyond the ablation in Table 2, we compare the baseline diffusion framework with its variant augmented by GSG.
> |Dataset|Method|$S_{\alpha}\uparrow$|$F_{\beta}^{\omega}\uparrow$|$E_{\phi}\uparrow$|$M\downarrow$|
> |:---:|:---:|:---:|:---:|:---:|:---:|
> |CAMO|-|.827|.797|.907|.056|
> |CAMO|+GSG|**.876**|**.852**|**.937**|**.038**|
> |COD10K|-|.827|.741|.932|.026|
> |COD10K|+GSG|**.859**|**.789**|**.934**|**.021**|
> |NC4K|-|.858|.823|.939|.034|
> |NC4K|+GSG|**.889**|**.859**|**.947**|**.026**|
>
> The results above show that GSG consistently improves all metrics across datasets.
>
> **Q2.**
> Our framework theoretically demonstrates the applicability of a one-step diffusion model for the COD task, under the premise of a properly designed model architecture. Accordingly, we introduce three key components (GSG, HCI, and L_RA) to enhance model performance. Furthermore, through experiments on a simple baseline, ablations with individual components, as well as the ablation experiment results reported in Tables 2, 3, and 4 of the main paper, we empirically validate the effectiveness of each component in improving COD performance.
> |Dataset|Method|$S_{\alpha}\uparrow$|$F_{\beta}^{\omega}\uparrow$|$E_{\phi}\uparrow$|$M\downarrow$|
> |:---:|:---:|:---:|:---:|:---:|:---:|
> |CAMO|-|.835|.806|.912|.053|
> |-|+HCI|.873|.846|.926|.041|
> |-|+GSG|.876|.852|.937|.038|
> |-|+LRA|.874|.854|.937|.040|
> |COD10K|-|.828|.741|.928|.026|
> |-|+HCI|.882|.826|.942|.017|
> |-|+GSG|.859|.789|.934|.021|
> |-|+LRA|.863|.798|.942|.020|
> |NC4K|-|.860|.824|.939|.034|
> |-|+HCI|.903|.879|.951|.022|
> |-|+GSG|.889|.859|.947|.026|
> |-|+LRA|.891|.864|.952|.025|
>
> **W1.** Since the CHAMELEON dataset contains only 76 images, reporting results on it may be less stable. To maintain consistency with prior diffusion-based methods, we did not include results on this dataset in the main paper. Additional experiments indicate that our method remains competitive on CHAMELEON.
> |Method|$S_{\alpha}\uparrow$|$F_{\beta}^{\omega}\uparrow$|$E_{\phi}\uparrow$|$M\downarrow$|
> |:---|:---:|:---:|:---:|:---:|
> |ZoomNeXt-P|**.924**|*.885*|.958|.018|
> |CamoDiffusion|.907|.875|**.961**|.020|
> |CODiff|*.914*|**.891**|*.960*|*.017*|
>
> **W2.** In Fig. 2, although the prediction (Pred) appears nearly identical to the ground truth (GT) visually, they are not the same image. This is because the model achieves very high accuracy on this sample, making the two almost indistinguishable at a visual level. We also confirm that there is an error in the “multiple” example shown in Fig. 3: one of the GT images was mistakenly mirrored during the figure preparation process. We will correct the issues in the final version.
>
> **W3.** Our method produces high-quality predictions via one-step inference and works reliably on most samples as shown in Figure 3 and Appendix E, including multiple scenarios. In very complex or strongly multi-modal cases, iterative diffusion may offer additional refinement. Overall, CODiff performs strongly in most cases, and we hope the community will continue exploring its limits.
>
> **W4.** Table 6 already reports GFLOPs and inference time per image compared to open-source diffusion-based COD methods. The results show that our method maintains competitive performance while being more efficient.
>
> **W5.** See **Q2**.
>
> **W6.** Following the reviewer’s suggestion, we have conducted additional comparisons with the relevant methods. Our model achieves SOTA performance on all three datasets; due to space limitations, we only report the results on the CAMO dataset. We will include a more comprehensive discussion of these methods in the final version.
> | Method    | $S_{\alpha} \uparrow$ | $F_{\beta}^{\omega} \uparrow$ | $M \downarrow$ | $E_{\phi} \uparrow$ |
> |:---------:|:-------------------:|:----------------------------:|:-------------:|:-----------------:|
> | FGSA-Net  | .889  | *.870*     | .036           | .944               |
> | SCOUT     | .859     | .919       | .047           | .828               |
> | RISE      | .722       | .587          | .113           | .775               |
> | EASE      | .749       | .684          | .098           | .831               |
> | UCOD-DPL  | .793   | .862         | .077           | .747               |
> | Ours      | **.894**    | **.881**    | **.033**       | **.947**           |

---

> > ### Author Rebuttal · Reviewer_B246 · 2026-04-04
> >
> > My concerns have been addressed. The authors' responses and additional evidence support my original rating.

---

> > > ### Author Response · Authors · 2026-04-06
> > >
> > > Dear Reviewer B246,
> > >
> > > Thank you for your constructive and insightful feedback, and for recognizing the contributions of CODiff. We sincerely appreciate the time and effort you devoted to reviewing our work. Your thoughtful comments have helped us further highlight the key merits of our method. We will incorporate the improvements into the final version.
> > >
> > > Best regards,
> > >
> > > The Authors

---

### Decision · Program_Chairs · 2026-04-30

**Decision:**

Accept (regular)

**Comment:**

This paper proposes CODiff, a one-step diffusion framework for camouflaged object detection. The work theoretically validates the feasibility of one-step sampling via probability flow ODEs, achieving state-of-the-art performance with significant efficiency gains.

Reviewers recognized the paper’s solid theoretical foundation, important innovation in applying the diffusion paradigm to camouflaged object detection, strong results on core benchmark datasets, and high-quality visualizations. They also noted limitations, including the omission of the CHAMELEON dataset, lack of comparisons with recent SOTA methods, insufficient analysis of efficiency and hard samples, and minor errors in figures.

In the rebuttal, the authors comprehensively addressed all reviewers’ concerns. They added missing experiments (CHAMELEON dataset, comparisons with recent baselines), corrected visualization issues, and refined theoretical derivations and quantitative analyses. These revisions notably improved the paper’s completeness and rigor, resolving all major and minor issues.

The initial inclination of the reviewers was weak accept. After their concerns were thoroughly addressed in the detailed rebuttal, and considering all feedback, I recommend that this paper be accepted to ICML 2026.